# Spatiotemporal dissection of the Golgi apparatus and the ER-Golgi intermediate compartment in budding yeast

**Takuro Tojima[1]\*, Yasuyuki Suda[1,2], Natsuko Jin[1], Kazuo Kurokawa[1], Akihiko Nakano[1]**

[1]Live Cell Super-Resolution Imaging Research Team, RIKEN Center for Advanced Photonics, Wako, Japan; [2]Laboratory of Molecular Cell Biology, Faculty of Medicine, University of Tsukuba, Tsukuba, Japan

**Abstract** Cargo traffic through the Golgi apparatus is mediated by cisternal maturation, but it remains largely unclear how the *cis*-cisternae, the earliest Golgi sub-compartment, is generated and how the Golgi matures into the *trans*-Golgi network (TGN). Here, we use high-speed and high-resolution confocal microscopy to analyze the spatiotemporal dynamics of a diverse set of proteins that reside in and around the Golgi in budding yeast. We find many mobile punctate structures that harbor yeast counterparts of mammalian endoplasmic reticulum (ER)-Golgi intermediate compartment (ERGIC) proteins, which we term 'yeast ERGIC'. It occasionally exhibits approach and contact behavior toward the ER exit sites and gradually matures into the *cis*-Golgi. Upon treatment with the Golgi-disrupting agent brefeldin A, the ERGIC proteins form larger aggregates corresponding to the Golgi entry core compartment in plants, while *cis*- and medial-Golgi proteins are absorbed into the ER. We further analyze the dynamics of several late Golgi proteins to better understand the Golgi-TGN transition. Together with our previous studies, we demonstrate a detailed spatiotemporal profile of the entire cisternal maturation process from the ERGIC to the Golgi and further to the TGN.

**\*For correspondence:**
takuro.tojima@riken.jp

**Competing interest:** The authors declare that no competing interests exist.

## Editor's evaluation

This paper provides important insights into the spatiotemporal mapping of a variety of proteins in and around the Golgi apparatus in budding yeast, with compelling evidence from high-resolution microscopy techniques. This research significantly advances our understanding of intracellular processes and represents a valuable contribution to the field of membrane trafficking and cell biology.

## Introduction

The Golgi apparatus plays a central role in membrane traffic (**Glick and Nakano, 2009**; **Nakano, 2022**; **Nakano and Luini, 2010**; **Pantazopoulou and Glick, 2019**). It consists of a series of flattened membrane sacs that can be classified into *cis*, medial, and *trans* cisternae. Newly synthesized cargo proteins that departed the endoplasmic reticulum (ER) enter the Golgi at the *cis* cisternae and move progressively through the medial and *trans* cisternae, and then proceed to the *trans*-Golgi network (TGN), a tubular-reticular membrane network that is joined to the *trans*-face of the Golgi (**Ford et al., 2021**; **Nakano, 2022**). This four-class classification of Golgi cisterna, i.e., *cis*, medial, *trans*, and TGN, is now widely used (sometimes the former two and the latter two are grouped together as early Golgi and late Golgi, respectively), but drawing clear boundaries between them is difficult because the

cargo traffic through the Golgi/TGN is mediated by cisternal maturation: an individual cisterna gradually changes its nature from an earlier to a later one, while the cargo proteins remain inside (*Casler et al., 2019*; *Kurokawa et al., 2019*; *Losev et al., 2006*; *Matsuura-Tokita et al., 2006*; *Tojima et al., 2019*). The Golgi cisternal maturation has been studied extensively in the budding yeast *Saccharomyces cerevisiae*. This is because, unlike other eukaryotic species, the Golgi in *S. cerevisiae* does not form stacks but exists as separate, individual cisternae scattered throughout the cytoplasm (*Beznoussenko et al., 2016*; *Preuss et al., 1992*; *Rambourg et al., 2001*), which gives us a great advantage in analyzing the dynamics of individual cisternae by live-cell imaging. Now, the next intriguing question is where and how *cis*-Golgi cisternae, the earliest sub-compartment of the Golgi maturation, is generated.

In mammalian cells, the ER-derived cargoes are carried by tubular-vesicular membrane clusters, termed the ER-Golgi intermediate compartment (ERGIC), to achieve long-distance cargo transport from the ER exit sites (ERES), distributed at the cell periphery, to the Golgi ribbon, a huge structure composed of interconnected Golgi stacks located near the centrosome. In plant cells, such long-distance transport is not necessary because separate Golgi ministacks are almost always in contact with the ERES (*Takagi et al., 2020*). We have recently identified the Golgi entry core compartment (GECCO), a specialized membrane compartment in plant cells, which appears to play similar roles to the mammalian ERGIC (*Ito et al., 2018*; *Ito et al., 2012*). The GECCO is located at the interface between the Golgi stack and ERES, and receives ER-derived cargoes for Golgi stack regeneration. However, it remains unclear whether such ER-Golgi intermediates (mammalian ERGIC or plant GECCO) also exist in budding yeast.

Another important issue that should be examined in budding yeast is the maturation process at the late Golgi. We have previously shown that, by live-cell imaging of a variety of late Golgi proteins, the yeast TGN can be classified into two distinct, sequential stages, early TGN and late TGN, which mediate cargo reception and export, respectively (*Tojima et al., 2019*). However, there are several additional important proteins that reside in the late Golgi but have not been well investigated.

In the present study, we address these challenges by the super-resolution confocal live imaging microscopy (SCLIM) that we developed (*Kurokawa and Nakano, 2020*; *Tojima et al., 2023*). Using fluorescent protein tags, we have investigated the spatiotemporal dynamics of the following 20 proteins involved in membrane traffic machinery in and around the Golgi: Emp46, a yeast counterpart of mammalian ERGIC-53 that acts as a cargo receptor for the transport of glycoproteins from the ER to the Golgi (*Sato and Nakano, 2002*); Ypt1, a yeast counterpart of mammalian Rab1 involved in ER-Golgi and endosome-Golgi traffic (*Jedd et al., 1995*; *Sclafani et al., 2010*; *Thomas et al., 2021*; *Thomas et al., 2018*); Rer1, a retrieval receptor for Golgi-to-ER retrograde traffic (*Sato et al., 1997*; *Sato et al., 2001*); Erd2, the K/HDEL-dependent retrieval receptor for Golgi-to-ER retrograde traffic (*Lewis et al., 1990*; *Semenza et al., 1990*), which has recently been proposed to reside in the Golgi and act as a gatekeeper without being recycled between ER and Golgi (*Alvim et al., 2023*); Grh1, a counterpart of the mammalian Golgi reassembly stacking protein of 65 kD (GRASP65) located at pre- or early-Golgi compartments (*Barr et al., 1997*; *Behnia et al., 2007*; *Kinseth et al., 2007*; *Tie et al., 2018*); Sed5, a target soluble *N*-ethylmaleimide-sensitive fusion attachment protein receptor (Qa-SNARE) involved in ER-Golgi and intra-Golgi traffic (*Hardwick et al., 1990*; *Nichols and Pelham, 1998*); Mnn9, a Golgi-resident mannosyltransferase in budding yeast used as a reference marker for *cis*-Golgi (*Ishii et al., 2016*); Mnn2, another Golgi-resident mannosyltransferase (*Orlean, 2012*; *Rayner and Munro, 1998*); Vrg4, a Golgi-resident mannose transporter (*Casler et al., 2019*; *Parker et al., 2019*); Sec21, a component (γ subunit) of the coat protein complex I (COPI) that mediates intra-Golgi and Golgi-ER retrograde traffic (*Ishii et al., 2016*; *Jackson, 2014*; *Papanikou et al., 2015*); Gnt1, a Golgi-resident glucose *N*-acetyltransferase used as a reference marker for medial-Golgi (*Ishii et al., 2016*; *Yoko-o et al., 2003*); Sys1, a Golgi membrane protein that recruits Imh1 (*Behnia et al., 2004*; *Tsukada and Gallwitz, 1996*), used as a reference marker for *trans*-Golgi (*Kurokawa et al., 2019*; *Tojima et al., 2019*); Imh1, the yeast Golgin protein that mediates endosome-Golgi traffic (*Chen et al., 2019*); Gea1 and Gea2, Golgi-resident guanine nucleotide exchange factors (GEFs) for adenosine diphosphate-ribosylation factor (Arf) mediating COPI vesicle formation (*Deng et al., 2009*; *Gustafson and Fromme, 2017*; *Peyroche et al., 2001*; *Spang et al., 2001*); Sec7, an Arf-GEF mediating carrier formation from the TGN (*Casanova, 2007*; *Richardson et al., 2012*), used as a reference marker for TGN (*Kurokawa et al., 2019*; *Tojima et al., 2019*); Tlg2, a Qa-SNARE mediating fusion

of endosome-derived vesicles with the TGN membrane (*Abeliovich et al., 1998*; *Chen et al., 2010*; *Holthuis et al., 1998*), used as an early TGN-specific marker (*Tojima et al., 2019*; *Toshima et al., 2023*); Apl6, a component (β subunit) of the clathrin-independent adaptor protein 3 (AP-3) complex involved in Golgi-vacuole traffic (*Cowles et al., 1997*; *Vowels and Payne, 1998*); Gga1, a Golgi-localized, γ-ear-containing, ARF-binding family protein (GGA) that acts as a clathrin adaptor at the TGN (*Black and Pelham, 2000*; *Zhdankina et al., 2001*); Ypt32, a yeast Rab11 homolog implicated in cargo export from the TGN (*Jedd et al., 1997*; *Suda et al., 2013*; *Thomas and Fromme, 2016*).

By combination of SCLIM observation and experiments using the Golgi-disrupting agent brefeldin A (BFA), we have identified the ERGIC in budding yeast and shown its approach and contact behavior toward the ERES and maturation into the Golgi. Furthermore, together with our previous studies (*Ishii et al., 2016*; *Suda et al., 2013*; *Tojima et al., 2019*), we have revealed an entire spatiotemporal profile of the cisternal maturation process from the ERGIC to the Golgi and then to the TGN.

## Results

### Identification of ERGIC in budding yeast

As a first step to examine whether the ERGIC exists in budding yeast, we focused on Emp46 (*Sato and Nakano, 2002*), a counterpart of mammalian ERGIC-53 that is widely used as a marker for ERGIC. A comparison of the amino acid sequences of human ERGIC-53 and yeast Emp46 (with an Emp46 paralog, Emp47) is shown in *Supplementary file 1*. We compared the spatiotemporal dynamics of EGFP-tagged Emp46 versus mCherry-tagged Mnn9, a mannnosyltransferase, in living yeast cells using SCLIM (*Figure 1A–E*; *Figure 1—figure supplement 1A, F*; *Table 1*). Mnn9 was used as a reference marker for *cis*-Golgi (*Ishii et al., 2016*). Four-dimensional (4D; xyz plus time) SCLIM imaging showed many mobile Mnn9-mCherry-positive and EGFP-Emp46-positive punctate compartments (~0.5 μm in diameter) appearing and disappearing with a lifetime of a few minutes. Dual-color analysis revealed that the punctate signals of EGFP-Emp46 appeared first and then Mnn9-mCherry signals showed up on the pre-existing EGFP-Emp46-positive compartment (*Figure 1B*). As the Mnn9-mCherry signal gradually increased, the EGFP-Emp46 signal decreased and eventually became invisible. The fluorescence intensity of EGFP-Emp46 peaked earlier than that of Mnn9-mCherry (*Figure 1C and D*). The transition from EGFP-Emp46 to Mnn9-mCherry proceeded by a gradual decrease of the area of the Emp46-positive zone and a complementary increase of the area of the Mnn9-positive zone. In many cisternae, the two zones often appeared to be spatially segregated, although some others showed overlapping distribution patterns (*Figure 1E*; *Figure 1—figure supplement 2*).

We also examined the dynamics of Ypt1 (the yeast counterpart of mammalian Rab1) versus Mnn9-mCherry (*Figure 1F–J*; *Figure 1—figure supplement 1B, F*; *Table 1*). In mammalian cells, Rab1 is involved in ER-Golgi cargo traffic (*Nuoffer et al., 1994*) and is used as another ERGIC marker (*Marie et al., 2009*; *Sannerud et al., 2006*). Using SCLIM, like Emp46, many mobile GFP-Ypt1-positive puncta were detected in a cell. Dual-color 4D analysis showed that the GFP-Ypt1 signal appeared first and then Mnn9-mCherry came up on the pre-existing GFP-Ypt1-positive compartment (*Figure 1G–I*). As the Mnn9-mCherry signal gradually increased, the GFP-Ypt1 signal decreased and eventually became invisible. Close-up images and their fluorescence line profiles (*Figure 1J*) showed that GFP-Ypt1- and Mnn9-mCherry-positive areas exhibit segregated distribution patterns during their transition period.

Next, we examined the dynamics of Golgi-resident retrieval receptors or gatekeepers for ER-resident proteins, Rer1 (*Sato et al., 1997*; *Sato et al., 2001*; *Figure 1K–N*; *Figure 1—figure supplement 1C, F*; *Table 1*) and Erd2 (*Alvim et al., 2023*; *Lewis et al., 1990*; *Semenza et al., 1990*; *Figure 1O–R*; *Figure 1—figure supplement 1D, F*; *Table 1*). Using SCLIM, we detected many mobile EGFP-Rer1- and Erd2-GFP-positive puncta in a cell. Dual-color 4D imaging showed that the peak time-points of EGFP-Rer1 and Erd2-GFP fluorescence signals came earlier than that of Mnn9-mCherry, consistent with the fact that ER-resident proteins are retrieved from the ER-Golgi intermediate compartment before reaching the Golgi (*Stornaiuolo et al., 2003*). We also observed that EGFP-Emp46 appeared at almost the same timing as Rer1-iRFP (*Figure 1S-V*; *Figure 1—figure supplement 1E, F*; *Table 1*). Taken together, these results indicate that Emp46, Ypt1, Rer1, and Erd2 appear earlier than Mnn9.

We further examined the dynamics of Grh1, a yeast homolog of the mammalian Golgi stacking protein GRASP65, which is located at the ER-Golgi interface in mammals (*Barr et al., 1997*; *Behnia et al., 2007*; *Kinseth et al., 2007*; *Tie et al., 2018*; *Figure 2*; *Figure 2—figure supplement 1*;

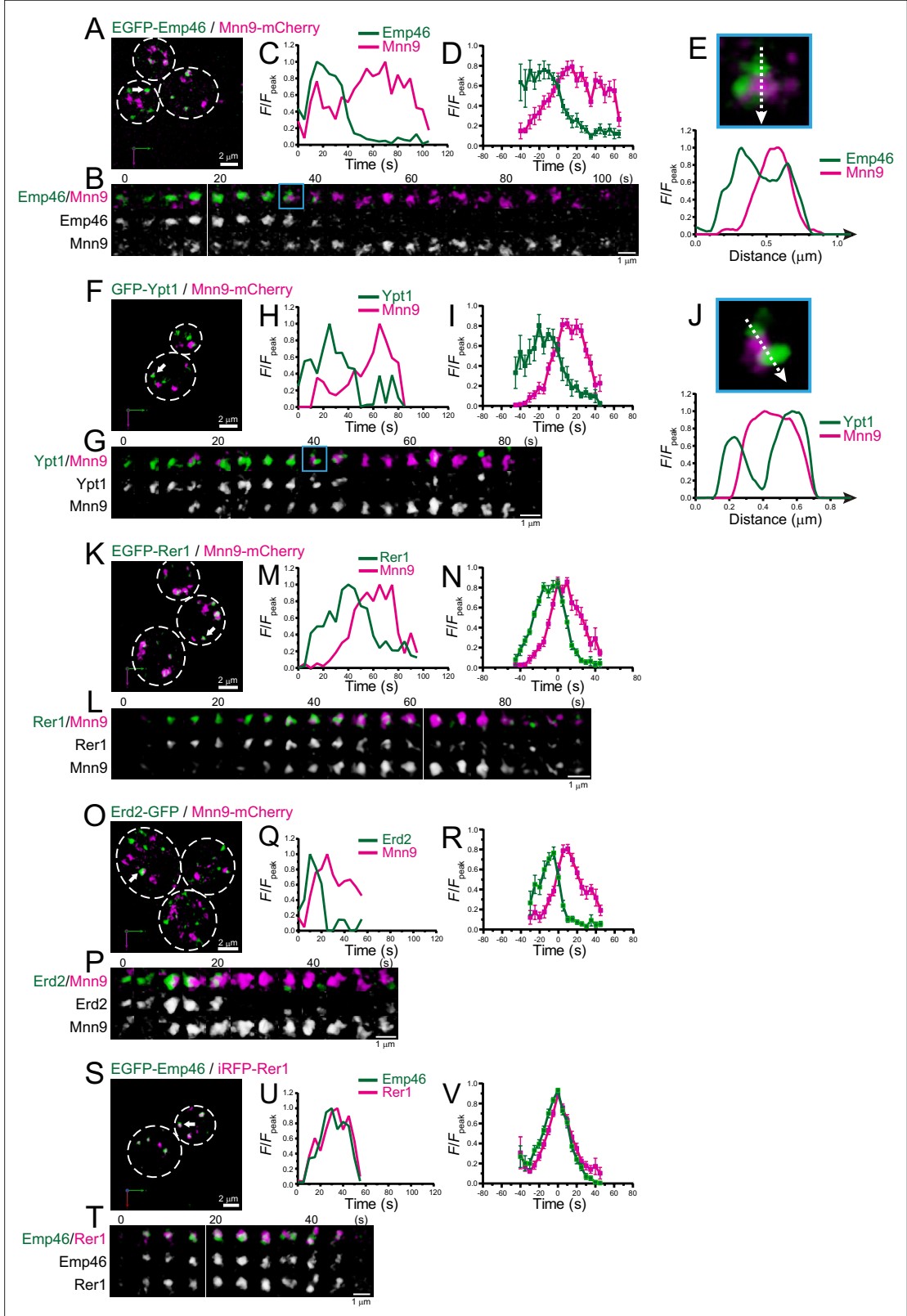

**Figure 1.** Four-dimensional (4D) dynamics of Emp46, Ypt1, Rer1, and Erd2. Dual-color 4D super-resolution confocal live imaging microscopy (SCLIM) imaging of yeast cells expressing EGFP-Emp46 and Mnn9-mCherry (**A–E**), GFP-Ypt1 and Mnn9-mCherry (**F–J**), EGFP-Rer1 and Mnn9-mCherry (**K–N**), Erd2-GFP and Mnn9-mCherry (**O–R**), and EGFP-Emp46 and Rer1-iRFP (**S–V**). (**A, F, K, O,** and **S**) Low-magnification images of the cells. The white broken lines indicate the edge of the cells. (**B, G, L, P,** and **T**) Time-lapse images of the single cisternae (white arrows in **A, F, K, O,** and **S**,

*Figure 1 continued on next page*

*Figure 1 continued*

respectively) in the cells. (**C, H, M, Q**, and **U**) Time course changes in relative fluorescence intensities ($F/F_{peak}$) of green and red channels in **B, F, J, M**, and **P**, respectively. (**D, I, N, R**, and **V**) Averaged time course changes in $F/F_{peak}$ of green and red channels (mean ± SEM). Time 0 was set as the midpoint between the green and red fluorescence peaks of each cisterna ($n$=10, 10, 14, 18, and 31 cisternae for **D, I, N, R**, and **V**, respectively). (**E** and **J**) Magnified images and their line scan analyses of EGFP-Emp46 and Mnn9-mCherry signals (**E**), and GFP-Ypt1 and Mnn9-mCherry signals (**J**) in the single maturing cisternae (blue rectangles in **B** and **G**). The $F/F_{peak}$ values (green and red channels) along the white broken lines in the upper panels are profiled in the graphs below. Scale bars: 2 μm (**A, F, K, O**, and **S**) and 1 μm (**B, G, L, P**, and **T**).

The online version of this article includes the following source data and figure supplement(s) for figure 1:

**Source data 1.** Data used for graphs presented in *Figure 1D, I, N, R, and V*.

**Figure supplement 1.** Individual data of fluorescence time courses and peak-to-peak times shown in *Figure 1*.

**Figure supplement 2.** Spatial distribution of Emp46 and Mnn9 within maturing cisterna.

*Table 1*). In budding yeast, Grh1-positive puncta were observed in the vicinity of the ERES (*Bruns et al., 2011*; *Levi et al., 2010*). Our dual-color 4D SCLIM imaging versus Mnn9 showed that the punctate Grh1-2xEGFP fluorescence signal first appeared and then changed into Mnn9-mCherry-positive compartment (*Figure 2A–D*; *Figure 2—figure supplement 1A, D*). We also found that Grh1-2xmCherry appeared before EGFP-Rer1 (*Figure 2E–H*; *Figure 2—figure supplement 1B, D*) and GFP-Ypt1 (*Figure 2I–L*; *Figure 2—figure supplement 1C, D*). These results indicate that Grh1 was the earliest protein to appear during cisternal maturation among those examined in our present and previous studies (*Ishii et al., 2016*; *Suda et al., 2013*; *Tojima et al., 2019*), supporting a hypothesis that Grh1 acts early in membrane traffic pathway (*Behnia et al., 2007*; *Levi et al., 2010*). At present, GRASP65 (Grh1 in yeast) is known as a typical *cis*-Golgi marker in mammals, but we find in the present study that Grh1 appears prior to the ERGIC markers Emp46 and Ypt1 in budding yeast (*Figures 1 and 2*). Therefore, we believe that Grh1 should be defined as another ERGIC marker rather than a *cis*-Golgi marker, at least in yeast. Together, our results suggest that the ERGIC exists in budding yeast and it matures into the *cis*-Golgi.

## 4D dynamics of early Golgi proteins

We performed dual-color 4D SCLIM imaging of several early Golgi proteins, Sed5 (Qa-SNARE), Vrg4 (mannose transporter), Sec21 (COPI component), and Mnn2 (mannosyltransferase), versus Mnn9 (*Figure 3*; *Figure 3—figure supplement 1*; *Table 1*). Sed5 and Vrg4 have been widely used as typical markers for *cis*-Golgi in budding yeast (*Kurokawa et al., 2014*; *Losev et al., 2006*; *Matsuura-Tokita et al., 2006*), as well as Mnn9. The time course of the punctate fluorescence signals of EGFP-Sed5 and GFP-Vrg4 were slightly different from that of Mnn9-mCherry: Sed5 appeared slightly earlier (*Figure 3A–D*; *Figure 3—figure supplement 1A, E*), whereas Vrg4 (*Figure 3E–H*; *Figure 3—figure supplement 1B, E*) appeared later, than Mnn9. Sec21-2xEGFP fluorescence appeared at almost the same time as Mnn9-mCherry, but the Sec21 signal increased slowly and peaked during the decay phase of Mnn9 (*Figure 3I–L*; *Figure 3—figure supplement 1C, E*). This is consistent with the notion that the COPI complex is located throughout the *cis*- to *trans*-Golgi cisternae. Mnn2-GFP signal initially appeared on the pre-existing Mnn9-mCherry-positive puncta and peaked during the decay phase of the Mnn9 signal (*Figure 3M–P*; *Figure 3—figure supplement 1D, E*), consistent with the fact that Mnn2 acts later than Mnn9 in the N-linked glycosylation pathway in budding yeast (*Orlean, 2012*).

## Subpopulation of ERGIC is located near the ERES

To investigate the spatial relationships between the ERGIC and ERES in budding yeast, we compared the localization of Grh1 or Rer1 versus Sec13, a component of the coat protein complex II (COPII) used as a marker for the ERES (*Kurokawa et al., 2014*; *Okamoto et al., 2012*; *Figure 4A–C*). Three-dimensional (3D; xyz) SCLIM imaging showed that the ERES labeled by Sec13-2xmCherry were distributed as many punctate signals (~0.5 μm in diameter). Using dual-color imaging, we found that some of the Grh1-2xGFP- and EGFP-Rer1-positive puncta were located adjacent to Sec13-2xmCherry-positive puncta, consistent with previous reports (*Bruns et al., 2011*; *Levi et al., 2010*). EGFP-Sed5-positive puncta were also found near Sec13-2xmCherry signals, although less frequently. In contrast, we could hardly detect such adjacent localization between Sec13-2xmCherry versus GFP-Vrg4 or Sec7-GFP. Quantitative correlation analysis (Pearson's *r*) indicated that the degree of co-localization

**Table 1.** Peak-to-peak duration time between the maximum fluorescence intensities of the two proteins at single cisterna.

| No. | Labeled proteins | Peak-to-peak duration (s) (mean ± SD) | *Number of puncta (n) in total cells (N) | | Figure No. |
|---|---|---|---|---|---|
| | | | n | N | |
| 1 | EGFP-Emp46 → Mnn9-mCherry | 31.0±24.1 | 10 | 6 | *Figure 1A–E*; *Figure 1—figure supplement 1A and F* |
| 2 | GFP-Ypt1 → Mnn9-mCherry | 29.5±14.6 | 10 | 8 | *Figure 1F–J*; *Figure 1—figure supplement 1B and F* |
| 3 | EGFP-Rer1 → Mnn9-mCherry | 13.2±11.7 | 14 | 9 | *Figure 1K–N*; *Figure 1—figure supplement 1C and F* |
| 4 | Erd2-GFP → Mnn9-mCherry | 16.4±10.3 | 18 | 14 | *Figure 1O–R*; *Figure 1—figure supplement 1D and F* |
| 5 | EGFP-Emp46 → iRFP-Rer1 | 3.9±9.0 | 31 | 15 | *Figure 1—figure supplement 1S–V*; *Figure 1—figure supplement 1E and F* |
| 6 | Grh1-2xEGFP → Mnn9-mCherry | 37.1±16.8 | 14 | 12 | *Figure 2A–D*; *Figure 2—figure supplement 1A and D* |
| 7 | Grh1-2xmCherry → EGFP-Rer1 | 26.2±17.7 | 13 | 9 | *Figure 2E–H*; *Figure 2—figure supplement 1B and D* |
| 8 | Grh1-2xmCherry → GFP-Ypt1 | 18.6±19.4 | 18 | 15 | *Figure 2I–L*; *Figure 2—figure supplement 1C and D* |
| 9 | EGFP-Sed5 → Mnn9-mCherry | 1.2±11.8 | 13 | 10 | *Figure 3A–D*; *Figure 3—figure supplement 1A and E* |
| 10 | Mnn9-mCherry → GFP-Vrg4 | 11.9±11.6 | 18 | 13 | *Figure 3E–H*; *Figure 3—figure supplement 1B and E* |
| 11 | Mnn9-mCherry → Sec21-2xEGFP | 37.9±33.2 | 17 | 13 | *Figure 3I–L*; *Figure 3—figure supplement 1C and E* |
| 12 | Mnn9-mCherry → Mnn2-GFP | 18.3±13.3 | 6 | 3 | *Figure 3M–P*; *Figure 3—figure supplement 1D and E* |
| 13 | Sys1-iRFP → GFP-Ypt1 | 19.7±16.3 | 18 | 13 | *Figure 6A–D*; *Figure 6—figure supplement 1A and C* |
| 14 | GFP-Ypt1 → Sec7-iRFP | 26.4±28.5 | 7 | 5 | *Figure 6E–H*; *Figure 6—figure supplement 1B and C* |
| 15 | Gea1-EGFP → Mnn9-mCherry | 6.0±10.0 | 15 | 9 | *Figure 7A–D*; *Figure 7—figure supplement 1A and F* |
| 16 | Mnn9-mCherry → Gea2-GFP | 76.2±21.7 | 13 | 11 | *Figure 7E–H*; *Figure 7—figure supplement 1B and F* |
| 17 | Gnt1-GFP → Gea2-2xmCherry | 30.0±12.4 | 15 | 12 | *Figure 7I–L*; *Figure 7—figure supplement 1C and F* |
| 18 | Gea2-2xmCherry → Tlg2-GFP | 9.3±11.0 | 21 | 11 | *Figure 7M–P*; *Figure 7—figure supplement 1D and F* |
| 19 | Gea2-GFP → Sec7-tagRFP | 23.6±16.8 | 7 | 6 | *Figure 7Q–T*; *Figure 7—figure supplement 1E and F* |
| 20 | Sys1-iRFP → Imh1-EGFP | 0.6±12.9 | 16 | 10 | *Figure 8A–D*; *Figure 8—figure supplement 1A and D* |
| 21 | Sec7-tagRFP → Gga1-GFP | 15.8±17.9 | 13 | 6 | *Figure 8E–H*; *Figure 8—figure supplement 1B and D* |
| 22 | Sys1-iRFP → Apl6-GFP | 5.8±27.8 | 13 | 11 | *Figure 8K–O*; *Figure 8—figure supplement 1C and D* |
| †23 | Sec7-mRFP → Ypt32-GFP | 51.9±23.2 | 7 | 4 | – |

*Number of puncta and cells used for the calculation of peak-to-peak duration. In each cell, 1–4 puncta were selected for the calculation.

†Suda, Y., Kurokawa, K., Hirata, R., and Nakano, A. 2013. Rab GAP cascade regulates dynamics of Ypt6 in the Golgi traffic. Proc. Natl. Acad. Sci. U.S.A. 110:18976–18981.

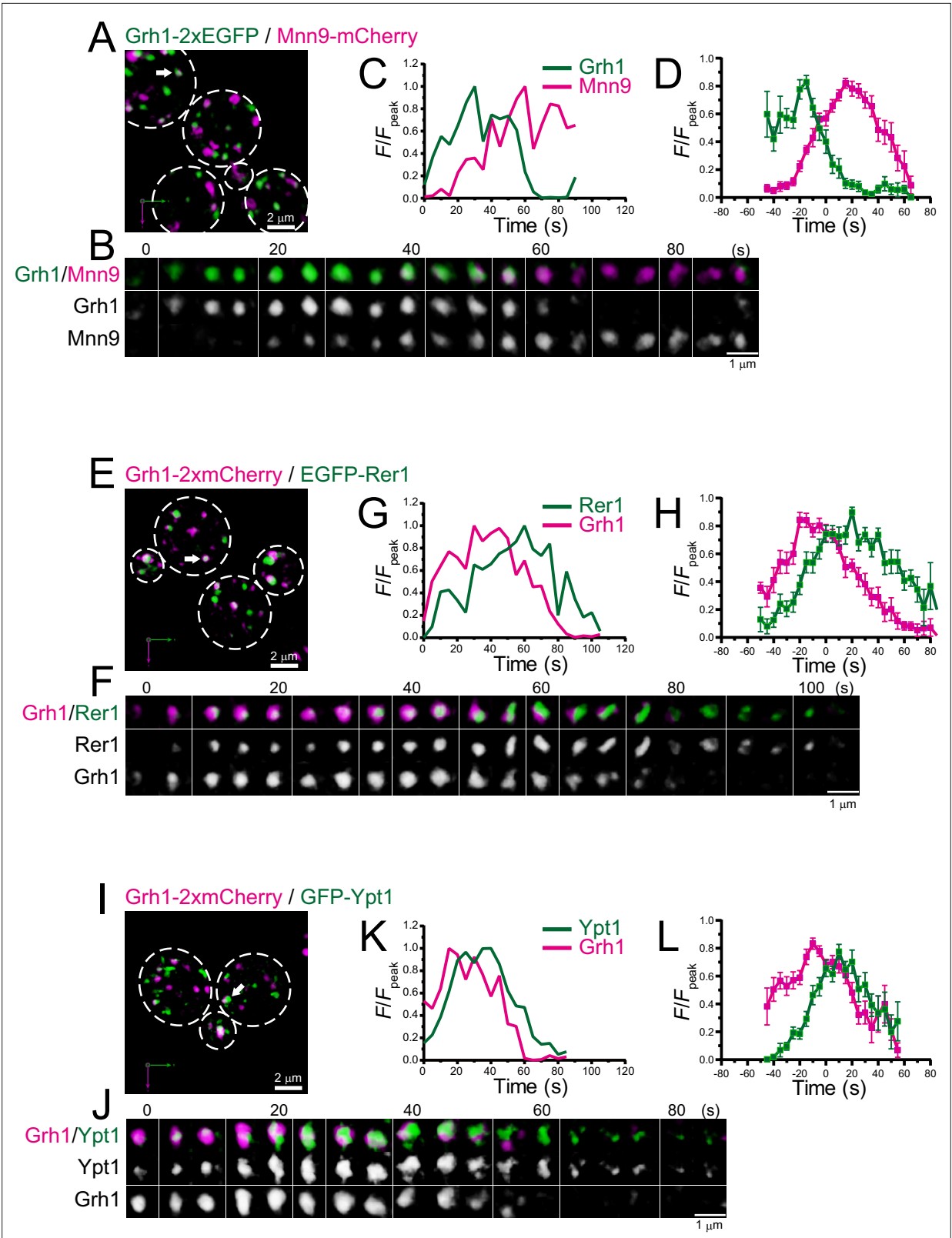

**Figure 2.** Four-dimensional (4D) dynamics of Grh1. Dual-color 4D super-resolution confocal live imaging microscopy (SCLIM) imaging of yeast cells expressing Grh1-2xEGFP and Mnn9-mCherry (**A–D**), Grh1-2xmCherry and EGFP-Rer1 (**E–H**), and Grh1-2xmCherry and GFP-Ypt1 (**I–L**). (**A, E**, and **I**) Low-magnification images of the cells. The white broken lines indicate the edge of the cells. (**B, F**, and **J**) Time-lapse images of the single cisternae (white arrows in **A, E**, and **I**, respectively) in the cells. (**C, G**, and **K**) Time course changes in relative fluorescence intensities ($F/F_{peak}$) of green and red channels in

*Figure 2 continued on next page*

*Figure 2 continued*

**B, F**, and **J**, respectively. (**D, H**, and **L**) Averaged time course changes in $F/F_{peak}$ of green and red channels (mean ± SEM). Time 0 was set as the midpoint between the green and red fluorescence peaks of each cisterna (*n*=14, 13, and 18 cisternae for **D, H**, and **L**, respectively). Scale bars: 2 µm (**A, E**, and **I**) and 1 µm (**B, F**, and **J**).

The online version of this article includes the following source data and figure supplement(s) for figure 2:

**Source data 1.** Data used for graphs presented in *Figure 2D, H, and L*.

**Figure supplement 1.** Individual data of fluorescence time courses and peak-to-peak times shown in *Figure 2*.

with Sec13 was relatively higher for Grh1 and Rer1, but moderate for Sed5 and lower for Vrg4 and Sec7 (*Figure 4C*), indicating that the earlier-appearing proteins during cisternal maturation tend to be located closer to the ERES. This suggests that the ERGIC is generated near the ERES, presumably by fusion of multiple ER-derived COPII vesicles carrying cargo. Alternatively, based on our previous finding that Sed5-positive cisterna exhibits approach and contact (hug-and-kiss) behavior toward the ERES to capture the cargo from the ER (*Kurokawa et al., 2014*), it is reasonable to speculate that Grh1- and Rer1-positive cisternae also exhibit the hug-and-kiss. To test this possibility, we performed dual-color 4D-SCLIM observations of Sec13-2xmCherry versus Grh1-2xEGFP (*Figure 4D*). We found that some Grh1-positive puncta (ERGIC) approached to Sec13-positive puncta (ERES) and temporarily associated with them for a certain period of time (2–3 s), and then dissociated. This looks exactly like the hug-and-kiss behavior we previously observed between *cis*-Golgi and ERES (*Kurokawa et al., 2014*).

## Yeast GECCO appeared by BFA treatment

We performed pharmacological experiments to dissect the ERGIC and Golgi in budding yeast (*Figure 5*). We used BFA, an antibiotic compound produced by a fungus, which is known as a potent inhibitor of GEFs for Arf GTPases (*Chardin and McCormick, 1999*; *Fujiwara et al., 1988*; *McCloud et al., 1995*). On the Golgi, Arf1 regulates the formation of COPI-coated vesicles that mediate retrograde transport from the Golgi to the ER (*Donaldson, 2005*). We have previously shown in plant cells that, upon BFA treatment, several Golgi marker proteins, ST (rat sialyltransferase lacking catalytic domain) and ATCASP (a putative Golgi matrix protein), were absorbed into the ER, whereas some pre/early-Golgi proteins, SYP31 (a homolog of yeast Sed5) and RER1B (a homolog of yeast Rer1), formed small punctate compartments in the vicinity of the ERES, which were named the 'GECCO' (*Ito et al., 2018*; *Ito et al., 2012*). After BFA removal, the ER-absorbed Golgi proteins first entered in, and then passed through, the GECCO to regenerate the Golgi stack. These observations strongly suggest that the plant GECCO is a specialized *cis*-Golgi sub-compartment acting as a cargo-entry site, which can be isolated upon BFA treatment. These characteristics of GECCO are similar to those of ERGIC in mammalian cells (*Ito and Boutté, 2020*; *Nakano, 2022*) (see Discussion for details).

Since wild-type strains of *S. cerevisiae* are insensitive to BFA, we used the *erg6* deletion mutant, which is defective in the biosynthesis of ergosterol, to make it sensitive to BFA (*Vogel et al., 1993*). We treated *erg6Δ* cells expressing Grh1-2xmCherry, EGFP-Rer1, EGFP-Sed5, GFP-Vrg4, or Mnn2-EGFP with dimethyl sulfoxide (DMSO; a vehicle control) (*Figure 5A*, upper panels) or BFA (*Figure 5A*, lower panels) for more than 60 min and observed by epifluorescence microscopy. All the DMSO-treated cells had many small punctate fluorescence signals (~0.5 µm in diameter) that correspond to the ERGIC or Golgi. In BFA-treated cells, the fluorescence signals of Grh1-2xmCherry, EGFP-Rer1, and EGFP-Sed5 were distributed in relatively larger and brighter aggregates (1–2 µm in diameter). On the other hand, GFP-Vrg4 and Mnn2-EGFP signals in BFA-treated cells were distributed in round and tubular structures that overlap with ER marker signals (mCherry-HDEL) (*Figure 5B*). We next performed dual-color 3D SCLIM observation (*Figure 5C and D*). In control cells (DMSO), small punctate signals of EGFP-Rer1 and Grh1-2xmCherry were partially segregated, which reflects the spatiotemporal status of cisternal maturation: i.e., if the timing of recruitment of the two proteins to a single cisterna is not perfectly synchronized, the cisterna would harbor only one of them at some time-points of maturation (see *Figure 2E–H*). In BFA-treated cells, however, the two signals mostly overlapped within larger aggregates (*Figure 5C*). Quantitative correlation analysis (Pearson's *r*) indicated that, compared to DMSO-treated cells, the degree of co-localization of Rer1 versus Grh1 was significantly higher in BFA-treated cells. Similar results were obtained when we compared the co-localization of

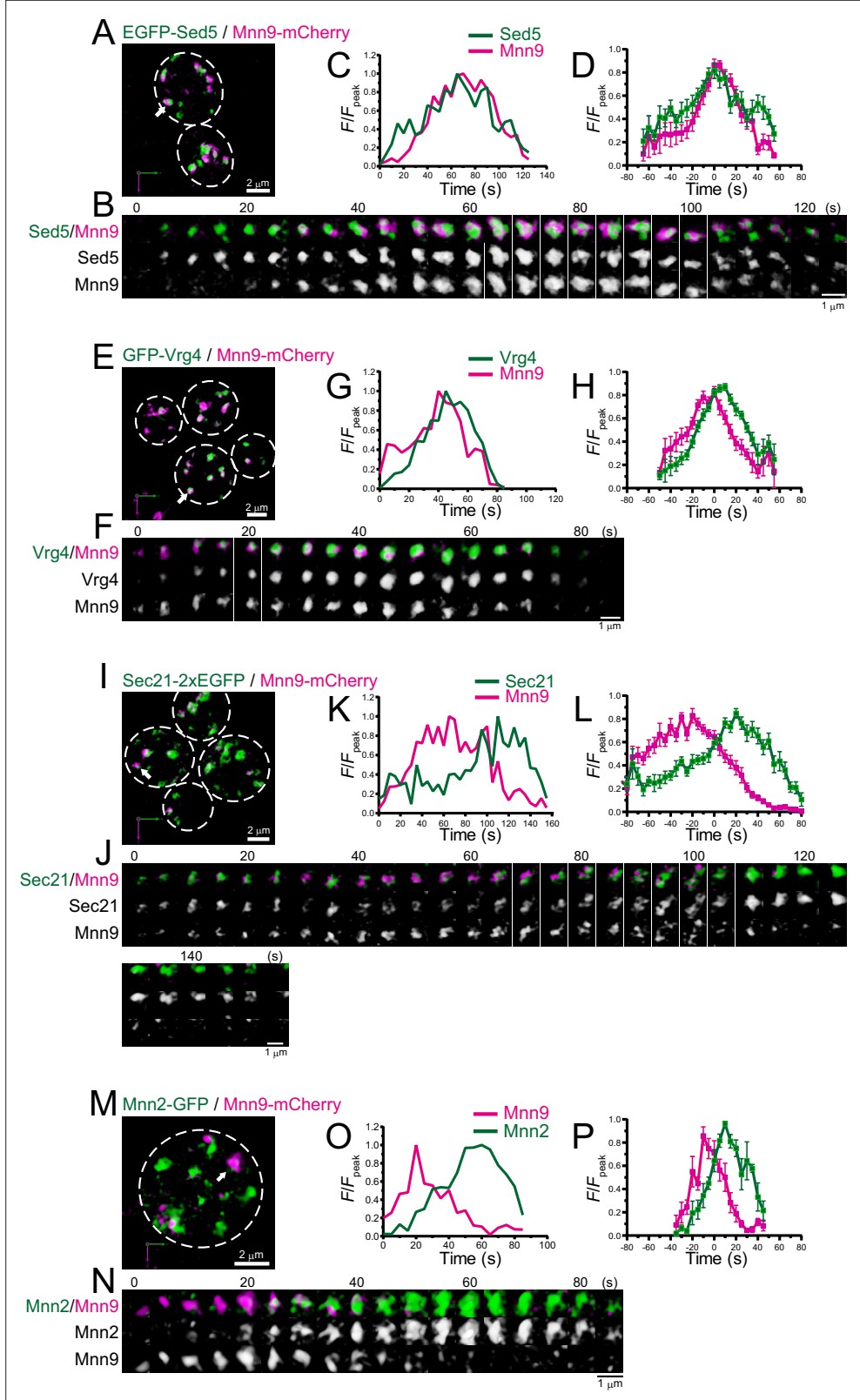

**Figure 3.** Four-dimensional (4D) dynamics of Sed5, Vrg4, Sec21, and Mnn2. Dual-color 4D super-resolution confocal live imaging microscopy (SCLIM) imaging of yeast cells expressing EGFP-Sed5 and Mnn9-mCherry (**A–D**), GFP-Vrg4 and Mnn9-mCherry (**E–H**), Sec21-2xEGFP and Mnn9-mCherry (**I–L**), and Mnn2-GFP and Mnn9-mCherry (**M–P**). (**A, E, I**, and **M**) Low-magnification images of the cells. The white broken lines indicate the edge

*Figure 3 continued on next page*

*Figure 3 continued*

of the cells. (**B, F, J**, and **N**) Time-lapse images of the single cisternae (white arrows in **A, E, I**, and **M**, respectively) in the cells. (**C, G, K**, and **O**) Time course changes in relative fluorescence intensities ($F/F_{peak}$) of green and red channels in **B, F, J**, and **N**, respectively. (**D, H, L**, and **P**) Averaged time course changes in $F/F_{peak}$ of green and red channels (mean ± SEM). Time 0 was set as the midpoint between the green and red fluorescence peaks of each cisterna ($n$=13, 18, 17, and 6 cisternae for **D, H, L**, and **P**, respectively). Scale bars: 2 μm (**A, E, I**, and **M**) and 1 μm (**B, F, J**, and **N**).

The online version of this article includes the following source data and figure supplement(s) for figure 3:

**Source data 1.** Data used for graphs presented in *Figure 3C, H, L, and P*.

**Figure supplement 1.** Individual data of fluorescence time courses and peak-to-peak times shown in *Figure 3*.

EGFP-Sed5 and Grh1-2xmCherry (*Figure 5D*). To confirm the effect of BFA on ERES, we performed 3D SCLIM observation of yeast cells expressing Sec13-mCherry (an ERES marker) and EGFP-Rer1 (*Figure 5—figure supplement 1*). Compared to DMSO-treated cells (vehicle control), the number of Sec13-positive puncta was unchanged in BFA-treated cells, whereas Rer1 formed larger aggregates, indicating that BFA had no effect on ERES.

We then examined time course and reversibility of the BFA effects by epifluorescence microscopy (*Figure 5E–G*). We perfused the cell chamber equipped on the microscope stage with DMSO and BFA sequentially and observed changes in the subcellular distribution of EGFP-Sed5 or GFP-Vrg4. Within 5 min after the change from DMSO to BFA, small punctate signals of EGFP-Sed5 turned to larger aggregates (*Figure 5F*, 10 min). Subsequently, within 5 min after BFA washout, the larger aggregates reverted to many small punctate structures (*Figure 5F*, 70 min). In contrast, upon perfusion treatment with BFA, GFP-Vrg4 signals initially became larger aggregates (*Figure 5G*, 10 min) and then were translocated to the ER network (*Figure 5G*, 20–60 min). After BFA washout, the Vrg4 tubular patterns gradually reverted to the original small punctate patterns (*Figure 5G*, 80–120 min).

Taken together, we have shown that, upon BFA treatment, the group of proteins that appear earlier during maturation (Grh1, Rer1, and Sed5) form larger aggregates, whereas the other group of proteins that appear later (Vrg4 and Mnn2) are absorbed into the ER. Since such BFA-induced protein redistribution in budding yeast is almost identical to that in plant cells (*Ito et al., 2018*; *Ito et al., 2012*), we hereafter refer to this BFA-induced aggregate as 'yeast GECCO' and propose to define Grh1, Rer1, Sed5, and presumably Emp46, as yeast ERGIC/GECCO components, although Sed5 was classified as a *cis*-Golgi protein in our previous studies (*Ishii et al., 2016*; *Kurokawa et al., 2014*).

## Dynamics of late Golgi proteins during maturation

Using SCLIM, we previously performed spatiotemporal mapping of 11 late Golgi proteins (Sec7, Clc1, Apl2, Gga2, Apl6, Chs5, Tlg2, Sys1, Ypt6, Sec21, and Gnt1), and showed that (1) the TGN is generated from the Golgi by cisternal maturation, (2) the TGN can be divided into two sequential stages, 'early TGN' and 'late TGN', and (3) Apl6, a component of the AP-3 complex, distributed as multiple small dot structures (<0.2 μm in diameter) on the TGN (*Tojima et al., 2019*). In the present study, using 4D SCLIM, we further investigated the dynamics of additional five late Golgi proteins, Ypt1, Gea2, Imh1, Gga1, and Ypt32, and re-analyzed Apl6 dynamics on the *trans*-Golgi, as described in the following subsections (*Figures 6–8*; *Table 1*).

## Ypt1 appears twice at ERGIC and late Golgi

We showed that Ypt1 appears at the ERGIC (*Figures 1 and 2*). In addition to its role in ER-Golgi traffic, Ypt1 has been implicated in endosome-Golgi traffic and vesicle formation at late Golgi (*McDonald and Fromme, 2014*; *Sclafani et al., 2010*). Indeed, a recent live-cell imaging analysis in budding yeast showed that Ypt1 transiently appears at late Golgi during maturation (*Thomas et al., 2018*). We performed dual-color 4D SCLIM imaging of GFP-Ypt1 versus Sys1-iRFP, a *trans*-Golgi marker, or Sec7-iRFP, a TGN marker (*Figure 6A–H*; *Figure 6—figure supplement 1*; *Table 1*). In multiple cases, the Ypt1 fluorescence signal appeared later than Sys1 and earlier than Sec7, suggesting that Ypt1 functions at the Golgi-TGN transition phase as well. To confirm that Ypt1 can localize at both ERGIC and *trans*/TGN cisterna, we performed triple-color 3D SCLIM imaging of GFP-Ypt1, Mnn9-mCherry, and Sec7-iRFP (*Figure 6I and J*). Here, Mnn9 and Sec7 were used as markers for *cis*-Golgi and the TGN, respectively. A subpopulation of Ypt1-positive puncta was located on Sec7-positive puncta

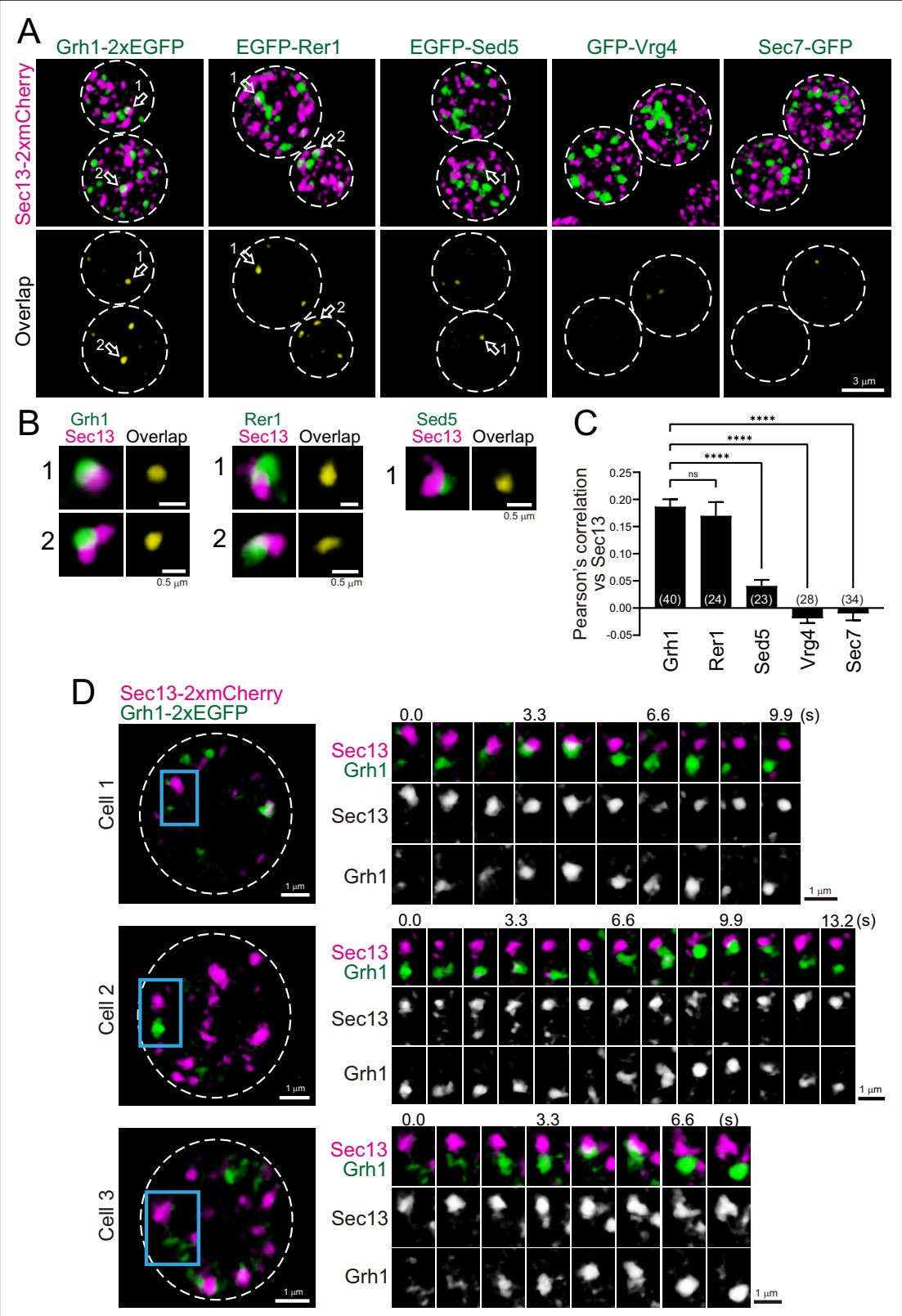

**Figure 4.** Three-dimensional (3D) distribution and 4D dynamics of endoplasmic reticulum exit sites (ERES) and ER-Golgi intermediate compartment (ERGIC) proteins. (**A**) Sec13-2xmCherry (ERES marker) was co-expressed with Grh1-2xEGFP, EGFP-Rer1, EGFP-Sed5, GFP-Vrg4, or Sec7-GFP and imaged by dual-color 3D super-resolution confocal live imaging microscopy (SCLIM). Upper panels show both red and green channels and lower panels show overlapping area only (yellow). Scale bar: 3 µm. (**B**) Zoom-up images of the overlapping areas of puncta indicated by arrows in **A**. Scale bars: 0.5 µm.

*Figure 4 continued on next page*

*Figure 4 continued*

(**C**) Pearson's correlation coefficient (*r*) for co-localization of the indicated proteins versus Sec13. Numbers in parentheses indicate the number of cells examined. Each value represents mean ± SEM. ns, not significant; ****p<0.0001, Dunnett's multiple comparison test. (**D**) Dual-color 4D SCLIM imaging of yeast cells expressing Sec13-2xmCherry and Grh1-2xEGFP. Three representative examples are shown (cells 1–3). The white broken lines in the left panels indicate the edge of the cells. The right panels show time-lapse images of the area covered by the blue rectangles in the left panels. Scale bars: 1 μm.

The online version of this article includes the following source data for figure 4:

**Source data 1.** Data used for the graph presented in *Figure 4C*.

(arrowheads in *Figure 6I*), while another was found on Mnn9-positive puncta (arrows in *Figure 6I*). The degrees of co-localization (Pearson's *r*) of Ypt1 versus Mnn9 and Ypt1 versus Sec7 were significantly higher than that of Mnn9 versus Sec7 (*Figure 6J*). These results strongly suggest that Ypt1 appears twice during the course of ERGIC-Golgi-TGN maturation.

## Gea2 appears at the interface between Golgi and TGN

There are at least three Arf-GEFs on the Golgi and TGN in budding yeast: Sec7, Gea1, and Gea2. While Sec7 is a typical marker for the TGN (*Tojima et al., 2019*), a recent study suggested that Gea1 and Gea2 are differentially distributed in the early and late Golgi, respectively (*Gustafson and Fromme, 2017*; *Highland and Fromme, 2021*). To further investigate the detailed spatiotemporal dynamics of Gea1 and Gea2, we performed dual-color 4D SCLIM imaging (*Figure 7*; *Figure 7—figure supplement 1*; *Table 1*). The time course of Gea1-EGFP fluorescence was nearly synchronized with that of Mnn9-mCherry, a *cis*-Golgi marker (*Figure 7A–D*; *Figure 7—figure supplement 1A, F*). In contrast, Gea2-GFP appeared much later than Mnn9-mCherry (*Figure 7E–H*; *Figure 7—figure supplement 1B, F*). Gea2-2xmCherry appeared later than Gnt1-GFP, a medial-Golgi marker (*Figure 7I–L*; *Figure 7—figure supplement 1C, F*), but earlier than GFP-Tlg2, an early TGN marker (*Figure 7M–P*; *Figure 7—figure supplement 1D, F*). In addition, we confirmed that Gea2-GFP appeared earlier than Sec7-tagRFP (*Figure 7Q–T*; *Figure 7—figure supplement 1E, F*), consistent with the previous report (*Highland and Fromme, 2021*). These results suggest that Gea2 functions at the interface between the Golgi and TGN.

## 4D dynamics of Imh1, Gga1, and Ypt32 at late Golgi

Imh1, a yeast GRIP domain-containing Golgin, has been implicated in endosome-Golgi traffic (*Wang et al., 2022*; *Yu and Lee, 2017*). We examined the dynamics of Imh1 versus Sys1, a *trans*-Golgi marker, using dual-color 4D SCLIM (*Figure 8A–D*; *Figure 8—figure supplement 1A, D*; *Table 1*). The time course of EGFP-Imh1 fluorescence was almost synchronized with that of Sys1-iRFP. This is consistent with a recent report showing that mammalian Sys1 recruits GRIP domain-containing Golgins (Golgin-245, Golgin-97, and GCC88) via ARFRP1 and ARL1, which capture retrograde transport carriers (*Ishida and Bonifacino, 2019*).

The GGA proteins are a family of monomeric adaptor proteins at the TGN. Two GGAs (Gga1 and Gga2) are present in budding yeast. We previously showed that Gga2 appeared transiently at Sec7-tagRFP-positive compartment during the decay phase of Sec7 (*Tojima et al., 2019*). We found that Gga1-GFP also appeared at Sec7-tagRFP-positive compartments during the decay phase of Sec7 (*Figure 8E–H*; *Figure 8—figure supplement 1B, D*; *Table 1*), implying functional redundancy of Gga1 and Gga2 in budding yeast.

In our previous study (*Suda et al., 2013*), we showed that Ypt32, the yeast counterpart of Rab11 that is involved in cargo export from the TGN, appears much later than Sec7. We compared the peak timing of Ypt32, Clc1, Gga1, Gga2, Apl2, and Chs5 relative to Sec7 (*Figure 9*; *Table 1*), and found that Ypt32 is the last protein to appear during TGN maturation among those examined in our present and previous studies (*Tojima et al., 2019*), consistent with a recent report (*Highland and Fromme, 2021*).

## AP-3 appears mainly at *trans*-Golgi

AP-3 is known to be involved in clathrin-independent traffic from the late Golgi to the vacuoles in yeast (*Cowles et al., 1997*; *Vowels and Payne, 1998*). Our previous 3D SCLIM analysis revealed that Apl6-positive punctate signals are segregated from Sec7-positive TGN compartments (*Tojima et al., 2019*). However, using 4D SCLIM analysis, we found that Apl6 was distributed as multiple

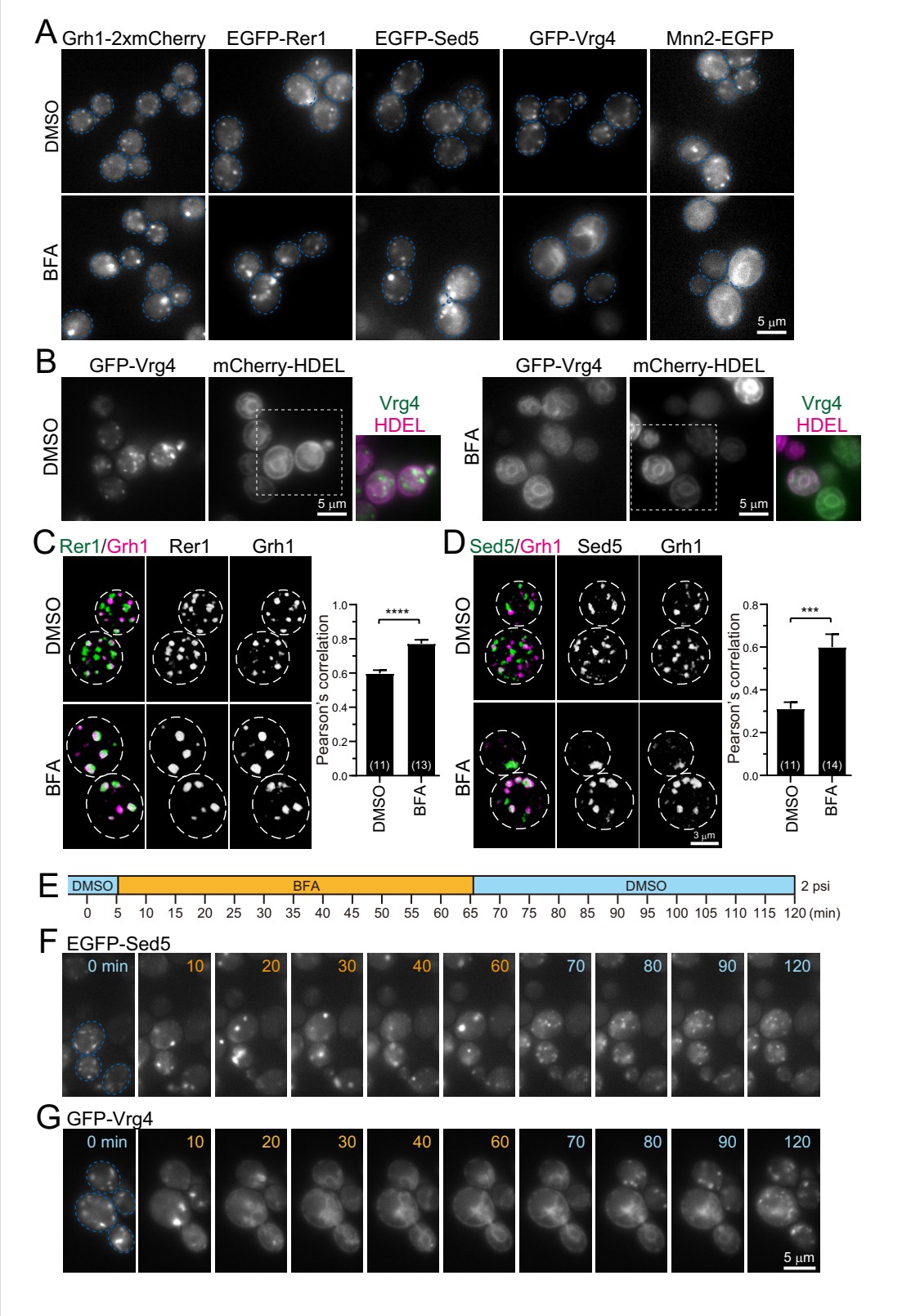

**Figure 5.** Effect of brefeldin A (BFA) on endoplasmic reticulum-Golgi intermediate compartment (ERGIC) and Golgi proteins. (**A**) Epifluorescence images of yeast cells expressing Grh1-2xmCherry, EGFP-Rer1, EGFP-Sed5, GFP-Vrg4, or Mnn2-EGFP treated with 0.4% dimethyl sulfoxide (DMSO) (control) (upper panels) or 200 μM BFA (lower panels). The blue broken lines indicate the edge of the cells. (**B**) Epifluorescence images of yeast cells co-expressing GFP-Vrg4 and mCherry-HDEL treated with 0.4% DMSO (left panels) or 200 μM BFA (right panels). Insets: merged images in the white dashed

*Figure 5 continued on next page*

*Figure 5 continued*

boxes. (**C** and **D**) Dual-color 3D SCLIM images of yeast cells co-expressing EGFP-Rer1 and Grh1-2xmCherry (**C**) or EGFP-Sed5 and Grh1-2xmCherry (**D**) treated with DMSO or BFA. Left, center, and right panels show merged, green, and red channels, respectively. The white broken lines indicate the edge of the cells. The graphs show Pearson's correlation coefficient (*r*) for co-localization. Numbers in parentheses indicate the number of cells examined. Each value represents mean ± SEM. ****p<0.0001; ***p<0.0005, unpaired *t*-test with Welch's correction. (**E–G**) BFA perfusion experiments. (**E**) Time schedule of drug perfusion. Time 0 indicates the onset of time-lapse imaging. The cells were originally exposed to 0.4% DMSO (light blue) solution in a perfusion chamber with a flow pressure of 2 psi. The perfusion solution was changed to 200 μM BFA (orange) at 5 min, and then returned to 0.4% DMSO (light blue) at 65 min to wash out BFA. (**F** and **G**) Time-lapse epifluorescence images of yeast cells expressing EGFP-Sed5 (**F**) or GFP-Vrg4 (**G**) treated sequentially with DMSO (0 min), BFA (10–60 min), and then DMSO again (70–120 min). The blue broken lines indicate the edge of the cells. Scale bars: 5 μm (**A, B**, and **G**) and 3 μm (**D**).

The online version of this article includes the following source data and figure supplement(s) for figure 5:

**Source data 1.** Data used for graphs presented in *Figure 5C, D*, and *Figure 5-supplement 1B*.

**Figure supplement 1.** Effect of brefeldin A (BFA) on endoplasmic reticulum exit sites (ERES).

small dot structures (<0.2 μm in diameter) on Sec7-positive compartments. Furthermore, we occasionally observed that Apl6-positive small dots already existed prior to the appearance of Sec7-positive compartments. These results suggest that AP-3 can assemble not only at the TGN but also at earlier Golgi stages. To test this possibility, we performed 3D co-localization analysis of Apl6 versus Mnn9, Sec21, Sys1, or Sec7 by dual-color SCLIM (*Figure 8I and J*). Here, Mnn9, Sec21, Sys1, and Sec7 were used as markers for *cis*-Golgi, *cis*-to-*trans*-Golgi, *trans*-Golgi, and TGN, respectively. The degree of co-localization of Apl6-GFP versus Sec21-2xmCherry and Sys1-iRFP was higher, whereas that versus Mnn9-mCherry and Sec7-tagRFP was lower, suggesting that AP-3 is mainly localized to the *trans*-Golgi rather than the TGN and *cis*-Golgi. We then performed dual-color 4D SCLIM to compare the spatiotemporal dynamics of Apl6 versus Sys1 (*Figure 8K–O*; *Figure 8—figure supplement 1C, D*; *Table 1*). Apl6-GFP signals initially appeared as multiple small dot structures (<0.2 μm in diameter) on and around the pre-existing Sys1-iRFP-positive compartment, and gradually increased in their volume to cover almost the half region of the compartment (*Figure 8L and O*; *Figure 8—figure supplement 2*). Subsequently, the accumulated Apl6-GFP signals gradually decreased in size and fragmented into many small dot-like structures on the Sys1-iRFP-positive compartment, and eventually became invisible. These results suggest that, unlike other adaptor proteins including adaptor protein 1 (AP-1), GGAs, and exomers, AP-3-mediated cargo export occurs at the *trans*-Golgi cisterna.

## ERGIC and ERES are distributed along with Golgi ribbon in mammalian cells

Finally, as a first step to compare the difference between budding yeast and mammalian cells, we observed the distribution of ERGIC and ERES in HeLa cells (*Figures 10 and 11*). Triple-color 3D SCLIM observation showed that majority of fluorescence signals of EGFP-ERGIC-53 (Emp46 in yeast) was distributed along with perinuclear Golgi ribbon (labeled by iRFP-ST), while many small punctate signals were also present throughout the cell (*Figure 10A-D*). Similarly, mScarlet-I-GRASP65 (Grh1 in yeast) and EGFP-Rab1 (Ypt1 in yeast) were distributed at the Golgi ribbon and small punctate compartments (*Figure 11*). We also confirmed that a large number of ERES (labeled by EGFP-Sec16B) were located adjacent to the Golgi ribbon as well as the cell periphery (*Figure 10E–H*). Moreover, ER tubular networks (labeled by mCherry-KDEL) were distributed throughout the cytoplasm including the vicinity of the Golgi ribbon (*Figure 10*). These observations indicate that the mammalian Golgi ribbon is surrounded by the ER, ERES, and ERGIC in close proximity, raising a possibility that the short-distance transport through the peri-Golgi ERES and ERGIC could be a dominant cargo transport pathway (*Saraste and Marie, 2018*).

## Discussion
### Definition of ERGIC

The ERGIC was originally identified in mammalian cells (*Saraste et al., 1987*; *Saraste and Svensson, 1991*; *Schweizer et al., 1988*; *Schweizer et al., 1990*) and considered as a carrier responsible for long-distance cargo transport from the ER at cell periphery to the Golgi near the nucleus (*Presley et al., 1997*; *Scales et al., 1997*). Its classical molecular markers are ERGIC-53 (Emp46 in yeast) and

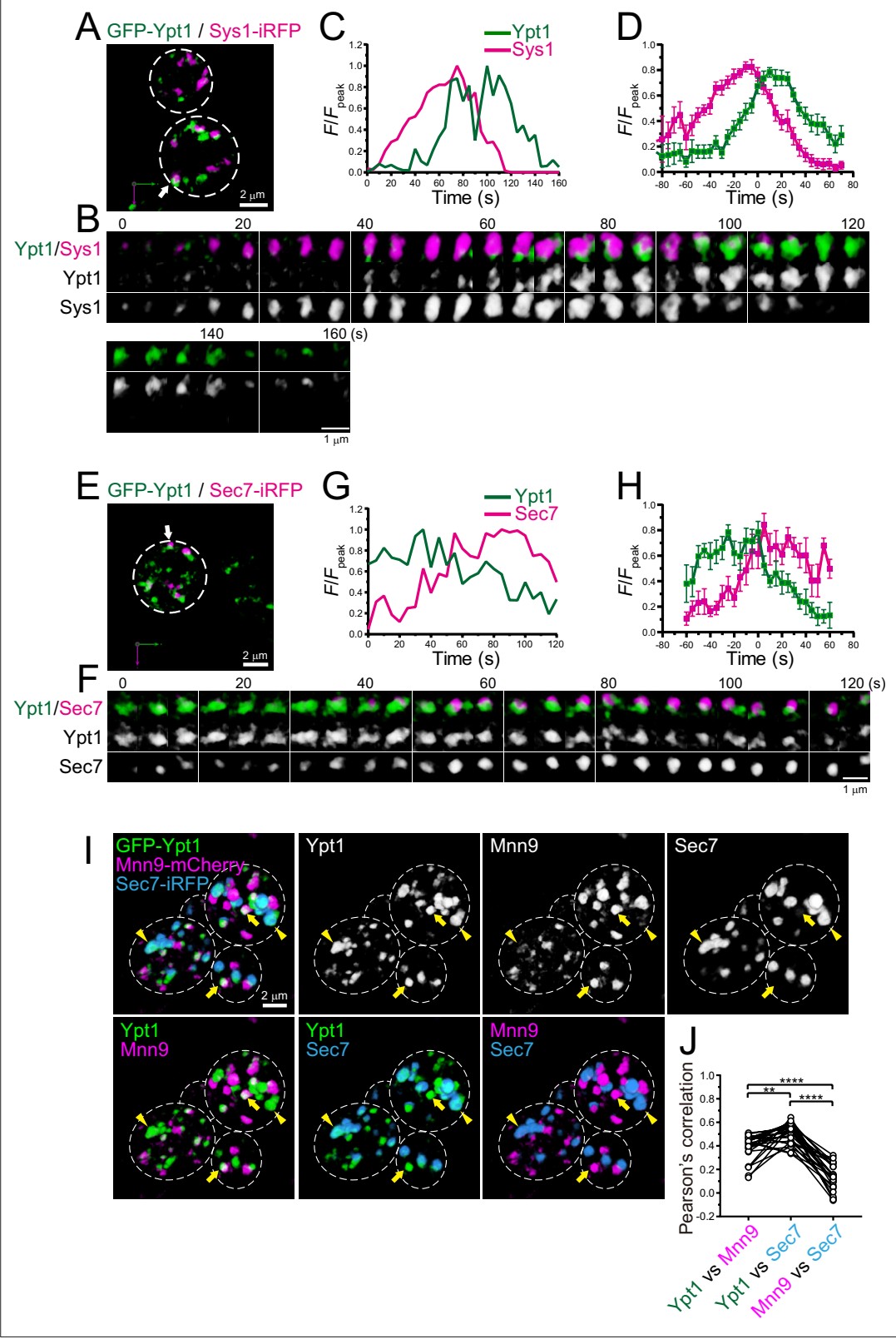

**Figure 6.** Four-dimensional (4D) dynamics and 3D distribution of Ypt1. (**A–H**) Dual-color 4D super-resolution confocal live imaging microscopy (SCLIM) imaging of yeast cells expressing GFP-Ypt1 and Sys1-iRFP (**A–D**), and GFP-Ypt1 and Sec7-iRFP (**E–H**). (**A** and **E**) Low-magnification images of the cells. The white broken lines indicate the edge of the cells. (**B** and **F**) Time-lapse images of the single cisternae (white arrows in **A** and **E**, respectively)

*Figure 6 continued on next page*

*Figure 6 continued*

in the cells. (**C** and **G**) Time course changes in relative fluorescence intensities ($F/F_{peak}$) of green and red channels in **B** and **F**, respectively. (**D** and **H**) Averaged time course changes in $F/F_{peak}$ of green and red channels (mean ± SEM). Time 0 was set as the midpoint between the green and red fluorescence peaks of each cisterna ($n$=18 and 7 cisternae for **D** and **H**, respectively). Scale bars: 2 μm (**A**, **E**) and 1 μm (**B**, **F**). (**I** and **J**) Triple-color 3D co-localization analyses of yeast cells expressing GFP-Ypt1, Mnn9-mCherry, and Sec7-iRFP. Upper left-most panel shows triple-merged image. Upper right three panels show individual single-channel images (Ypt1, Mnn9, and Sec7). Lower panels show dual-merged images. Yellow arrows indicate cisternae that contain Ypt1 and Mnn9, but not Sec7. Yellow arrowheads indicate cisternae that contain Ypt1 and Sec7, but not Mnn9. (**J**) Pearson's correlation coefficient ($r$) for co-localization of the indicated pair of proteins. **p<0.01; ****p<0.0001, Tukey's multiple comparison test ($n$=25 cells).

The online version of this article includes the following source data and figure supplement(s) for figure 6:

**Source data 1.** Data used for graphs presented in *Figure 6D, H, and J*.

**Figure supplement 1.** Individual data of fluorescence time courses and peak-to-peak times shown in *Figure 6*.

Rab1 (Ypt1 in yeast), and structurally, vesicular-tubular clusters containing COPII and COPI coats are thought to be a feature of ERGIC (*Aridor et al., 1995*; *Peter et al., 1993*; *Saraste and Svensson, 1991*; *Schweizer et al., 1988*). Thus, the ER-ERGIC cargo traffic has been believed to be mediated by COPII vesicles budded and released from the ERES. However, a recent sensational study by correlative light and electron microscopy challenges this conventional view and has shown that ERES located at the cell periphery extend long tubules carrying cargo along microtubules toward a perinuclear Golgi ribbon in HeLa cells (*Weigel et al., 2021*). This work suggests that ERGIC can be initially generated as a specialized membrane domain at ERES.

In plants, ERES and Golgi stack are almost always adjacent to each other (*Takagi et al., 2020*), so there is no need for such a long-distance carrier as in mammals. We have previously shown that the *cis*-most cisterna of the plant Golgi has very similar properties to the mammalian ERGIC and is deemed a functionally specialized compartment for cargo reception, which we named GECCO (*Ito et al., 2018*; *Ito et al., 2012*). In mammalian cells, microtubule disruption leads to the disassembly of the Golgi ribbon and the formation of many Golgi ministacks in the vicinity to the ERES (*Cole et al., 1996*; *Hammond and Glick, 2000*). In such situations, the ERGIC components are sandwiched between the Golgi and ERES (*Tie et al., 2018*), very much like the case of the plant Golgi. Even when the microtubule network is intact, substantial amounts of ERGIC markers are observed at the *cis*-face of the perinuclear Golgi ribbon (present study, *Figures 10 and 11*; *Ladinsky et al., 1999*; *Mellman and Simons, 1992*). Furthermore, a recent report by super-resolution imaging has classified mammalian ERGIC into two sub-compartments according to their spatial localization: Golgi-associated ERGIC, which is attached to the *cis*-face of the Golgi ribbon, and peripheral ERGIC, which is separate from the Golgi ribbon (*Wong-Dilworth et al., 2023*). A subpopulation of ERES is also distributed in the vicinity of the Golgi ribbon (present study, *Figure 10*; *Hammond and Glick, 2000*). These facts suggest that the short distance transport through the ERES near the Golgi ribbon and the Golgi-associated ERGIC is a cargo transport mechanism conserved between plants and mammals (*Saraste and Marie, 2018*).

One other amazing feature about ERGIC/GECCO is the insensitivity to the Arf-GEF inhibitor BFA. Most of Golgi-resident proteins are absorbed into the ER upon treatment with BFA, while ERGIC components remain independent as will be discussed in more detail later. This property of ERGIC appears to be common to plant and yeast cells as well as animal cells (even in *Drosophila* cells, see *Fujii et al., 2020*).

To summarize, the new pre-Golgi compartment we have found in yeast in the present study well corresponds to the concept of mammalian ERGIC and plant GECCO in terms of molecular components and functions, so we consider that they are homologous compartments. Since each Golgi cisterna in yeast moves around in the cytoplasm away from the ERES, we feel that the new yeast compartment is more in line with the original concept of mammalian ERGIC, which is based on the mobility as an ER-Golgi carrier. Therefore, we use the term 'yeast ERGIC' in this paper and propose that the major functions and molecular components of the ERGIC/GECCO are evolutionarily conserved across species, even though their subcellular distribution looks different.

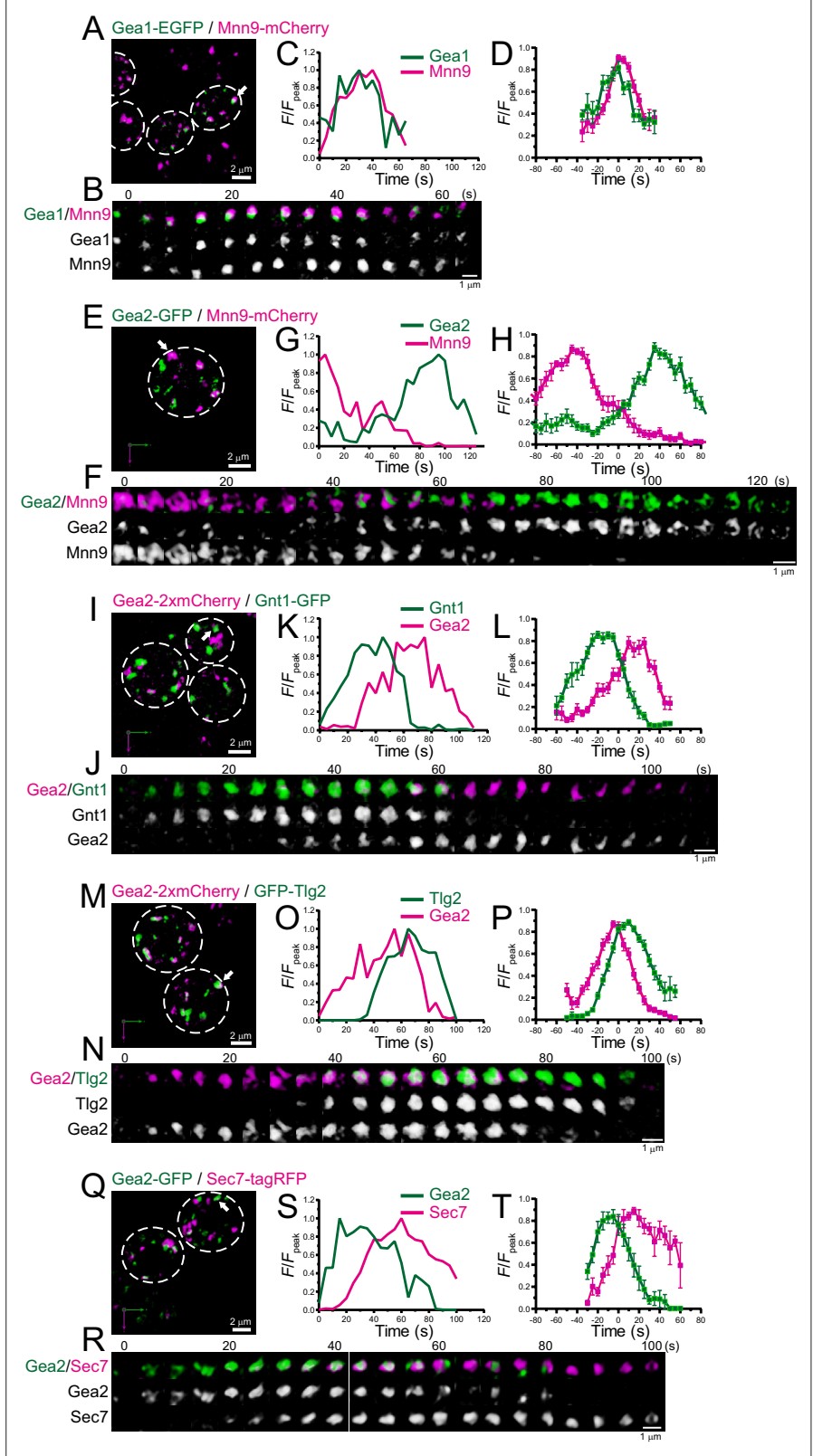

**Figure 7.** Four-dimensional (4D) dynamics of Gea1 and Gea2. Dual-color 4D super-resolution confocal live imaging microscopy (SCLIM) imaging of yeast cells expressing Gea1-EGFP and Mnn9-mCherry (**A–D**), Gea2-GFP and Mnn9-mCherry (**E–H**), Gea2-2xmCherry and Gnt1-GFP (**I–L**), Gea2-2xmCherry and GFP-Tlg2 (**M–P**), and Gea2-GFP and Sec7-tagRFP (**Q–T**). (**A, E, I, M**, and **Q**) Low-magnification images of the cells. The white broken lines indicate

*Figure 7 continued on next page*

*Figure 7 continued*

the edge of the cells. (**B, F, J, N**, and **R**) Time-lapse images of the single cisternae (white arrows in **A, E, I, M**, and **Q**, respectively) in the cells. (**C, G, K, O**, and **S**) Time course changes in relative fluorescence intensities ($F/F_{peak}$) of green and red channels in **B, F, J, N**, and **R**, respectively. (**D, H, L, P**, and **T**) Averaged time course changes in $F/F_{peak}$ of green and red channels (mean ± SEM). Time 0 was set as the midpoint between the green and red fluorescence peaks of each cisterna (*n*=15, 13, 15, 21, and 7 cisternae for **D, H, L, P**, and **T**, respectively). Scale bars: 2 µm (**A, E, I, M**, and **Q**) and 1 µm (**B, F, J, N**, and **R**).

The online version of this article includes the following source data and figure supplement(s) for figure 7:

**Source data 1.** Data used for graphs presented in *Figure 7D, H, L, P, and T*.

**Figure supplement 1.** Individual data of fluorescence time courses and peak-to-peak times shown in *Figure 7*.

## ERGIC matures into Golgi in yeast

The ERGIC in budding yeast, which we have identified in the present study, matures into the Golgi (*Figures 1 and 2*). To date, it has been debated whether the mammalian ERGIC represents a transient ER-Golgi transport carrier or a stable pre-Golgi organelle (*Saraste and Marie, 2018*). Our results support the former model, at least in budding yeast. Intriguingly, our SCLIM technology revealed that, during the ERGIC-Golgi transition, the early and late markers are often spatially separated in a single cisterna, forming a boundary between two stages. Similar segregation patterns were also observed during the Golgi-TGN maturation process by SCLIM (*Tojima et al., 2019*). Our previous correlative light and electron microscopy analysis showed that Golgi and TGN markers are segregated within a continuous membrane compartment (*Kurokawa et al., 2019*). We propose that the ERGIC and Golgi, as well as Golgi and TGN, can transiently coexist as structurally and functionally distinct zones within a single, maturing cisterna. The molecular mechanisms of this segregation will be a challenging problem to be addressed in the future.

At present, it remains unclear whether the mammalian and plant Golgi are also generated from ERGIC/GECCO by maturation. Higher spatiotemporal resolution live imaging is necessary and our second-generation SCLIM technology will be quite helpful (*Tojima et al., 2023*; Miyashiro, Tojima, and Nakano, in revision). We consider that cisternal maturation is not a mechanism specific to the yeast Golgi, but rather a universal cargo transfer mechanism that is frequently used in many membrane traffic pathways in diverse organisms. Consistent with this idea, maturation of early endosomes into late endosomes has also been demonstrated in mammalian cells (*Huotari and Helenius, 2011*; *Rink et al., 2005*).

## Effects of BFA on plant, yeast, and mammalian cells

In plant cells, we have previously identified the GECCO as a *cis*-most Golgi compartment that receives cargoes from the ER, which becomes evident upon BFA treatment. BFA caused the accumulation of a specialized set of proteins located at the *cis*-most cisternae of the Golgi stack, such as SYP31 (Sed5 in yeast) and RER1B (Rer1 in yeast), into small punctate structures (GECCO) (*Ito et al., 2018*; *Ito et al., 2012*). On the other hand, other Golgi proteins such as ST were excluded from the GECCO and absorbed into the ER. After BFA removal, the GECCO received the absorbed Golgi proteins from the ER to regenerate the Golgi stack. These results suggest that the GECCO contains functional components to receive cargoes from the ER. This property is very similar to that of mammalian ERGIC (*Ito et al., 2018*).

When we treated budding yeast cells with BFA in the present study, Grh1, Rer1, and Sed5 accumulated into larger punctate structures (yeast GECCO), whereas Mnn2 and Vrg4 were absorbed into the ER (*Figure 5A and B*). A notable difference between plant and yeast GECCO is their size: the yeast GECCO is larger (1–2 µm in diameter) than the plant GECCO (~0.5 µm in diameter). Time course observation of plant GECCO formation showed that, within 20 min after BFA application, SYP31 temporally congregated and coalesced into several large aggregates (2–10 µm in diameter), and then the aggregates gradually disassembled into many small punctate structures near the ERES (*Ito et al., 2012*). On the other hand, Sed5 in budding yeast formed large aggregates soon after BFA treatment (<5 min), but did not lead to their subsequent disassembly (*Figure 5F*). Considering that the plant Golgi forms stacks and are tethered to the ERES, whereas the yeast Golgi cisternae are unstacked and majority of them moves independently of the ERES, we assume that the disassembly of the plant

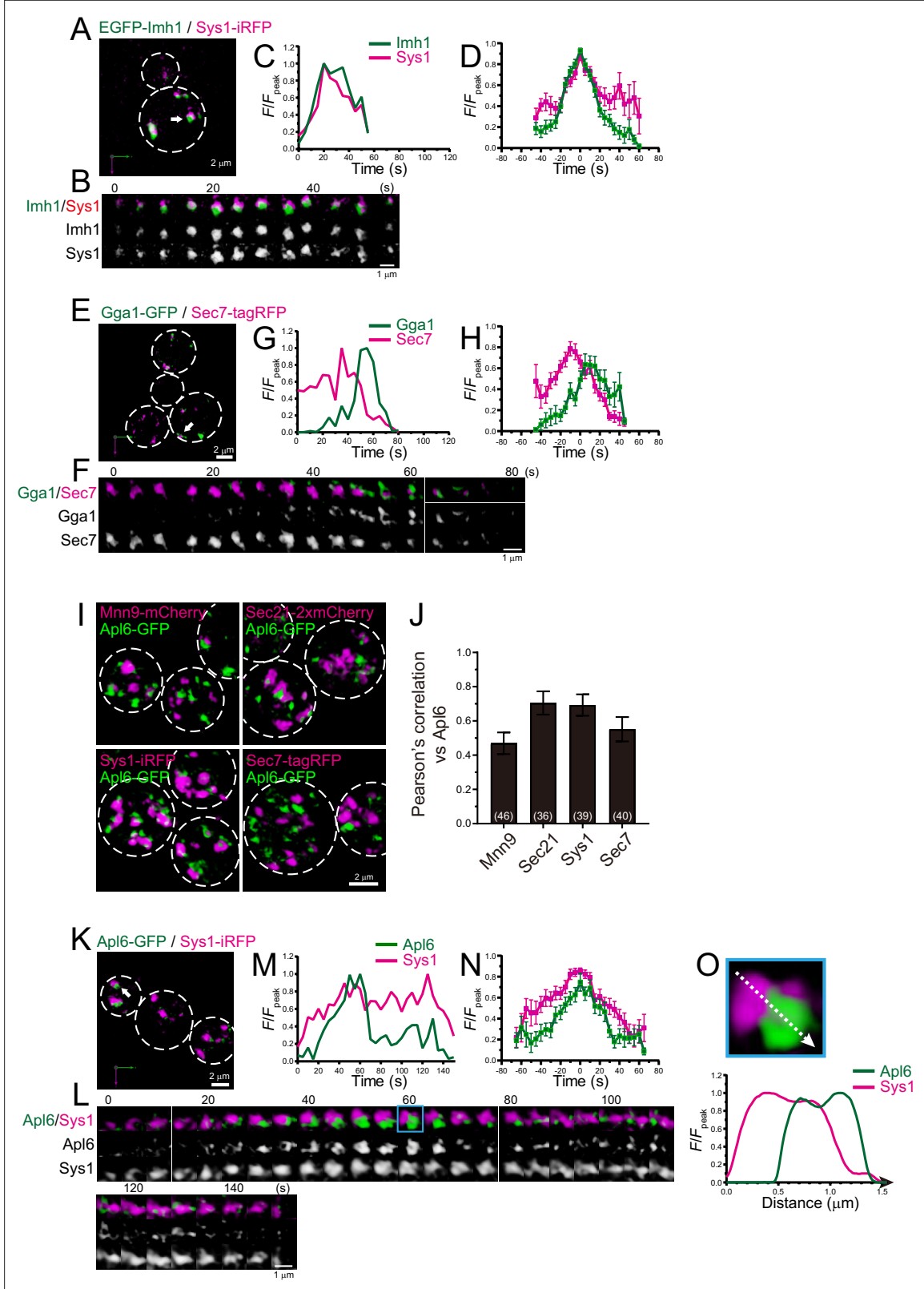

**Figure 8.** Four-dimensional (4D) dynamics and 3D distribution of Imh1, Gga1, and Apl6. (**A–H**) Dual-color 4D super-resolution confocal live imaging microscopy (SCLIM) imaging of yeast cells expressing EGFP-Imh1 and Sys1-iRFP (**A–D**), and Gga1-GFP and Sec7-tagRFP (**E–H**). (**A** and **E**) Low-magnification images of the cells. The white broken lines indicate the edge of the cells. (**B** and **F**) Time-lapse images of the single cisternae (white arrows in **A** and **D**, respectively) in the cells. (**C** and **G**) Time course changes in relative fluorescence intensities (*F/F*<sub>peak</sub>) of green and red channels in

*Figure 8 continued on next page*

*Figure 8 continued*

B and E, respectively. (D and H) Averaged time course changes in $F/F_{peak}$ of green and red channels (mean ± SEM). Time 0 was set as the midpoint between the green and red fluorescence peaks of each cisterna (*n*=16 and 13 cisternae for D and H, respectively). (I and J) 3D co-localization analyses of Apl6 versus Golgi/*trans*-Golgi network (TGN) marker proteins. (I) GFP-Apl6 was co-expressed with Mnn9-mCherry (*cis*-Golgi), Sec21-2xmCherry (*cis*/medial-Golgi), Sys1-iRFP (*trans*-Golgi), and Sec7-tagRFP (TGN) and imaged by dual-color 3D SCLIM. (J) Pearson's correlation coefficient (*r*) for co-localization of the indicated proteins versus Apl6. Numbers in parentheses indicate the number of cells examined. Each value represents mean ± SD. (K–O) Dual-color time-lapse SCLIM imaging of yeast cells expressing Apl6-GFP and Sys1-iRFP. (K) Low-magnification image of the cells. The white broken lines indicate the edge of the cells. (L) Time-lapse images of the single cisterna (white arrows in K) in the cell. (M) Time course changes in relative fluorescence intensities ($F/F_{peak}$) of green and red channels in L. (N) Averaged time course changes in $F/F_{peak}$ of green and red channels (mean ± SEM). Time 0 was set as the midpoint between the green and red fluorescence peaks of each cisterna (*n*=13 cisternae). (O) Magnified image and line scan analyses of Apl6-GFP and Sys1-iRFP signals in the single maturing cisterna (blue rectangle in L). The $F/F_{peak}$ values (green and red channels) along the white broken lines in the upper panels are profiled in the graphs below. Scale bars: 2 µm (A, E, I, and L) and 1 µm (B, F, and L).

The online version of this article includes the following source data and figure supplement(s) for figure 8:

**Source data 1.** Data used for graphs presented in *Figure 8D, H, J, and N*.

**Figure supplement 1.** Individual data of fluorescence time courses and peak-to-peak times shown in *Figure 8*.

**Figure supplement 2.** Spatial distribution of Apl6 and Sys1 within maturing cisterna.

GECCO is regulated by putative ER-Golgi-tethering molecules, the presence of which is suggested by laser trapping experiments (*Sparkes et al., 2018*).

In mammalian cells, previous reports have shown that BFA treatment reversibly induces Golgi disassembly and absorption of many Golgi proteins into the ER (*Barzilay et al., 2005*; *Scheel et al., 1997*). However, the ERGIC marker ERGIC-53 (Emp46 in yeast) localizes to punctate structures near the ERES, termed 'Golgi remnants' (*Lippincott-Schwartz et al., 1990*). Similarly, several other pre/early-Golgi proteins, including GRASP65 (Grh1 in yeast), syntaxin 5 (Sed5 in yeast), and RER1, form punctate structures located adjacent to the ERES (*Füllekrug et al., 1997*; *Nakamura et al., 1995*; *Seemann et al., 2000*; *Tang et al., 1993*; *Ward et al., 2001*). These results strongly suggest that the GECCO observed in plants and budding yeast are equivalent to the Golgi remnant in mammalian cells.

## Dynamics of ERES and ERGIC in yeast

The yeast Golgi is unstacked and individual Golgi cisternae move through the cytoplasm in a spatially separated state from the ERES. It has long been thought that *cis*-Golgi compartment is generated near the ERES, presumably by fusion of multiple ER-derived COPII vesicles carrying cargo. In contrast, we recently found in yeast that the *cis*-Golgi compartment exhibits approach and contact (hug-and-kiss) behavior toward the ERES to capture cargo (*Kurokawa et al., 2014*). In the present study, we found the presence of ERGIC as a pre-*cis*-Golgi compartment, and it also exhibits the hug-and-kiss (*Figure 4D*). Considering that ERGIC had a higher probability of localizing near the ERES than *cis*-Golgi (*Figure 4A–C*), it is most likely that yeast ERGIC exhibits the hug-and-kiss behavior more frequently than *cis*-Golgi. The next intriguing questions would be where and how the ERGIC compartment is generated, and whether the molecular mechanisms of ERGIC formation and cargo loading are shared.

## Ypt1 appears twice during cisternal maturation

The functions of Ypt1 at the Golgi have been controversial so far. Earlier studies showed that Ypt1 regulates the ER-Golgi transport (*Jedd et al., 1995*). Subsequent studies have demonstrated additional roles for Ypt1 at the late Golgi, including endosome-Golgi transport and cargo export from the TGN (*McDonald and Fromme, 2014*; *Sclafani et al., 2010*; *Thomas et al., 2018*). In the present study, we revealed that Ypt1 appeared twice during cisternal maturation, at the ERGIC and at the *trans*-Golgi/TGN interface (*Figures 2I–L, 6A–H, and 9*). This suggest that Ypt1 is involved in multiple functions at the different stages of Golgi maturation. In contrast to Ypt1 in budding yeast, Rab1 in mammalian cells has been reported to localize only to the ERGIC/*cis*-Golgi (*Tie et al., 2018*). Given that the yeast genome contains only 11 Rab proteins, whereas the human genome contains at least 60 (*Homma et al., 2021*; *Pfeffer, 2017*), we can speculate that a single Rab protein is evolved to execute multiple functions at multiple stages of the Golgi maturation in budding yeast.

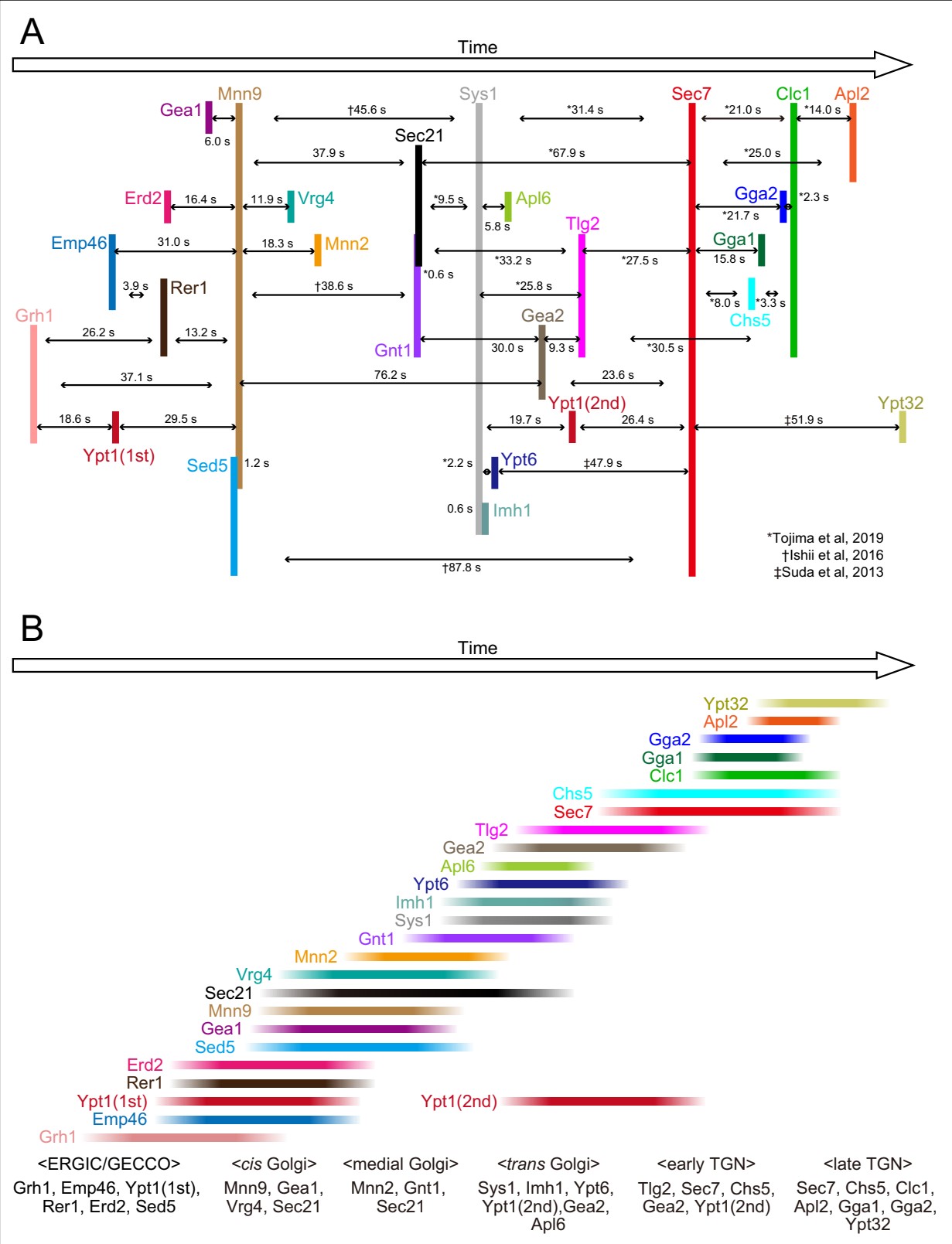

**Figure 9.** Temporal order of all the proteins examined in the present and previous studies. (**A**) Peak-to-peak duration times obtained in our previous (*Ishii et al., 2016*; *Suda et al., 2013*; *Tojima et al., 2019*) and present studies. Time flows from left to right. The lengths of the black horizontal lines with arrows reflect the peak-to-peak times (s). (**B**) The colored lines show the residence periods of the indicated proteins at a single ERGIC/Golgi/*trans*-Golgi network (TGN) cisterna. We propose that the endoplasmic reticulum Golgi intermediate compartment (ERGIC)-Golgi maturation process can be

*Figure 9 continued on next page*

*Figure 9 continued*

divided into following successive stages: the 'ERGIC/Golgi entry core compartment (GECCO)' stage, which is defined by the presence of Grh1, Emp46, Ypt1 (first appearance), Rer1, Erd2, and Sed5, the 'cis-Golgi' stage, defined by the presence of Mnn9, Gea1, Vrg4, and Sec21, the 'medial-Golgi' stage, defined by the presence of Mnn2, Gnt1, and Sec21, and the 'trans-Golgi' stage, defined by the presence of Sys1, Imh1, Ypt6, Ypt1 (second appearance), Gea2, and Apl6. In addition, we previously defined the 'early TGN stage' and the 'late TGN stage' (*Tojima et al., 2019*).

## Differential localization and functions of Gea1 and Gea2

Previous studies have shown that the Arf1-GFFs Gea1 and Gea2 orchestrate the formation of COPI vesicles destined for the ER and earlier Golgi compartments (*Peyroche et al., 2001*; *Spang et al., 2001*). It has been suggested that Gea1 and Gea2 have overlapping functions, since deletion of either gene results in no detectable phenotype whereas the double null mutation is lethal (*Spang et al., 2001*). However, recent studies have revealed that Gea1 and Gea2 are differentially distributed in the early and late Golgi, respectively (*Gustafson and Fromme, 2017*; *Highland and Fromme, 2021*). In the present study, we performed the detailed temporal mapping of Gea1 and Gea2 during maturation, and found that Gea1 appeared at the *cis*-Golgi, whereas Gea2 appeared at the interface between the *trans*-Golgi and early TGN (*Figure 7*). Gea1 and Gea2 have been reported to be recruited by Ypt1 (*Gustafson and Fromme, 2017*), which is consistent with our observation that the timing of the first and second recruitments of Ypt1 were close to the appearance of Gea1 and Gea2, respectively (*Figure 9*). Furthermore, the appearance of Gea2 at late Golgi is consistent with a report showing that Gea2 forms a complex with the small GTPase Arl1 and the lipid flippase Drs2 to recruit the Golgin Imh1 to the late Golgi, which is involved in endosome-Golgi traffic (*Yu and Lee, 2017*). Our observation that Imh1 appeared around the *trans*-Golgi (*Figure 8A–D*) also supports the involvement of Gea2 for this process. Taken together, Gea1 and Gea2 likely have distinct roles at different stages of Golgi maturation, although they share several molecular interactors and essential functions.

## AP-3 mainly localizes at the *trans*-Golgi

We have previously shown that, in budding yeast, many coat and adaptor proteins that mediate cargo export, i.e., clathrin (Clc1), AP-1 (Apl2), GGA2 (Gga2), and exomer (Chs5), assemble at the TGN (*Tojima et al., 2019*). Furthermore, in the present study, we showed that GGA1 (Gga1) assembled at the TGN with similar dynamics as GGA2 (*Figure 8E–H*). In contrast, only a small fraction of the AP-3 (Apl6) was observed at the TGN, but the majority was located at the *trans*-Golgi (*Figure 8I–O*). A previous study also showed that Apl5, another AP-3 component, appears prior to Sec7, Apl2, and Gga2 (*Highland and Fromme, 2021*). These results suggest that AP-3 mediates cargo transport from the *trans*-Golgi, but not from the TGN, to the vacuole. This is consistent with the notion that the AP-3-mediated cargo export is clathrin-independent (*Cowles et al., 1997*; *Vowels and Payne, 1998*), because clathrin appears at the late TGN but not at the *trans*-Golgi (*Tojima et al., 2019*). Intriguingly, our SCLIM imaging showed that Apl6 signals at the *trans*-Golgi covered a partial region of the cisterna (*Figure 8L and O*; *Figure 8—figure supplement 2*), suggesting the formation of cargo export/sorting zones at the *trans*-Golgi. This is reminiscent of our previous study showing that the plant TGN also has discrete zones responsible for distinct cargo sorting: AP-1 and AP-4 zones (*Shimizu et al., 2021*).

## The classification of Golgi maturation stages

Based on our previous (*Tojima et al., 2019*) and present studies, we here attempt to update the previous definitions and molecular markers for each Golgi stage in budding yeast. We propose that the Golgi maturation process, from ERGIC to TGN, can be divided into at least five functional stages: (1) the 'ERGIC/GECCO' stage, which mediates ER-Golgi traffic; (2) the 'cis- and medial-Golgi' stage, which mediates carbohydrate synthesis; (3) the 'trans-Golgi' stage, which serves as a scaffold for TGN assembly; (4) the 'early TGN' stage, which mediates cargo entry into the TGN; and (5) the 'late TGN' stage, which mediates cargo export from the TGN (*Figure 9*). Now that we define the ERGIC/GECCO stage in budding yeast, several proteins (including Sed5) that were previously categorized as *cis*-Golgi proteins should now be redefined as ERGIC proteins. Although the appearance period of Sed5 largely overlaps with that of *cis*-Golgi proteins such as Mnn9 (*Figure 3A–D*), the ERGIC and *cis*-Golgi may coexist as spatially distinct zones within a cisterna. There may also be an argument about the role of *cis*-Golgi, because carbohydrate synthesis was attributed to the medial- and *trans*-Golgi but not the *cis*-most Golgi in mammalian and plant cells (*Day et al., 2013*). Giving a new functional criterion to

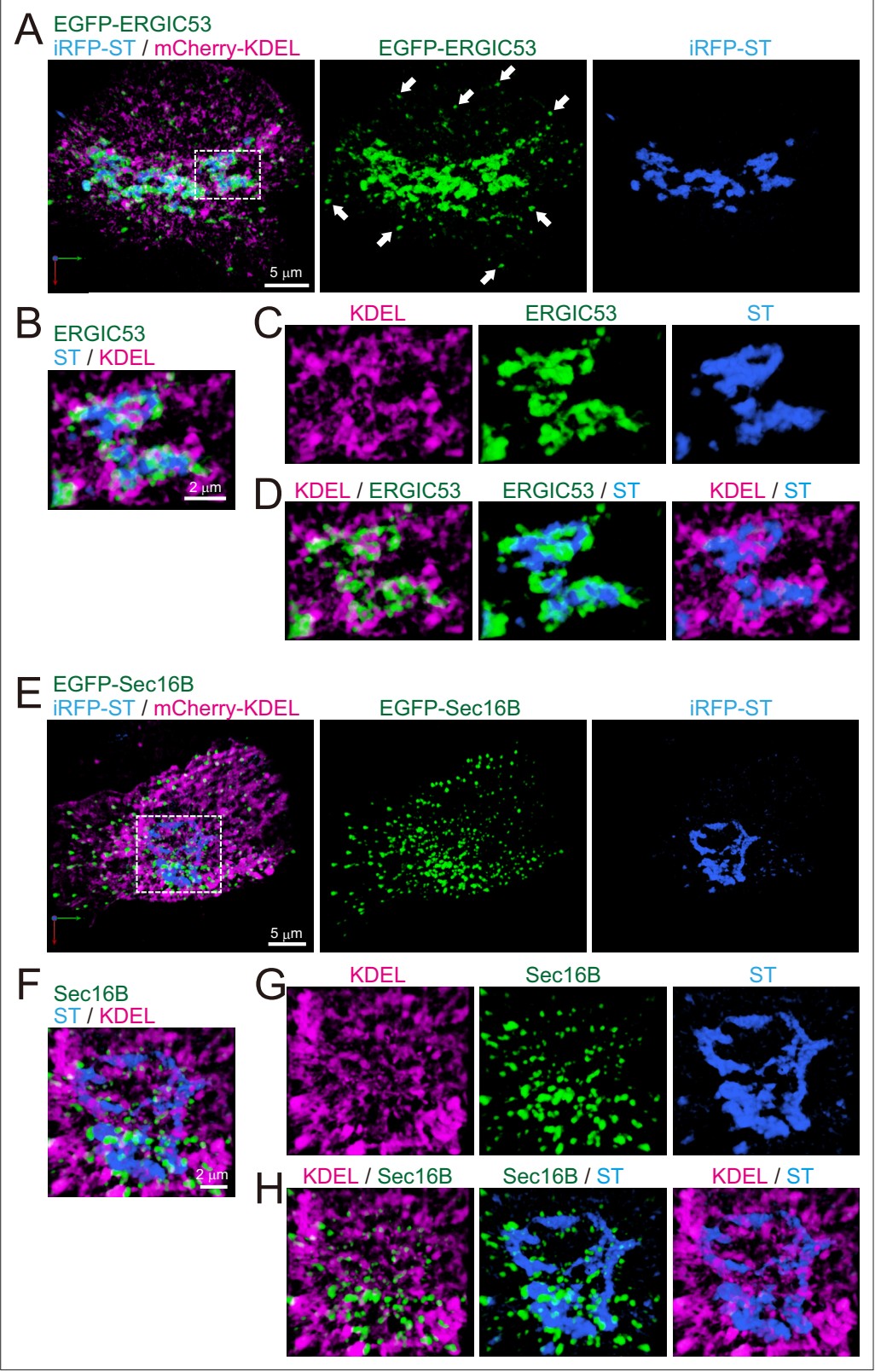

**Figure 10.** Distribution of endoplasmic reticulum-Golgi intermediate compartment (ERGIC), ER exit sites (ERES), Golgi, and ER in HeLa cells. (**A–D**) Triple-color three-dimensional (3D) super-resolution confocal live imaging microscopy (SCLIM) images of the Golgi ribbon area of a HeLa cell expressing EGFP-ERGIC53 (ERGIC marker, green), iRFP-ST (Golgi marker, blue), and mCherry-KDEL (ER marker, magenta). (**E–H**) Triple-color 3D SCLIM

*Figure 10 continued on next page*

*Figure 10 continued*

images of the Golgi ribbon area of a HeLa cell expressing EGFP-Sec16B (ERES marker, green), iRFP-ST (blue), and mCherry-KDEL (magenta). (**A** and **E**) Images of the whole Golgi ribbon. The left panels show merged images. The center and right panels show EGFP and iRFP fluorescence signals, respectively. (**B–D** and **F–H**) Zoom-up images of a portion of the Golgi ribbon in the white dashed squares shown in (**A**) and (**E**). (**B** and **F**) Triple-merged images. (**C** and **G**) Single-channel images. (**D** and **H**) Dual-merged images. Arrows in (**A**): EGFP-ERGIC53-positive puncta distributed away from the Golgi ribbon. Scale bars: 5 µm (**A** and **E**) and 2 µm (**B** and **F**).

the *trans*-Golgi as a scaffold for the TGN assembly is also new, but it is important to consider this role at the Golgi-TGN boundary because there are drastic differences in functions and origins between the Golgi and the TGN (*Nakano, 2022*). The temporal order of the appearance of 25 proteins we examined in budding yeast (*Figure 9*) is well comparable to the spatial order of localization of more than 50 proteins within the Golgi stack in mammalian cells (*Tie et al., 2018*), suggesting a universality of Golgi components, organization, and dynamics across species. Understanding of this universality of Golgi will help comprehend the fundamental basis of Golgi function, structure, and dynamics revealed in many species.

In conclusion, the present study has revealed the dynamic nature of Golgi generation and maturation, in which a wide variety of functional molecules move in and out of the cisternae in a spatio-temporally controlled manner. Furthermore, the identification of ERGIC in budding yeast suggests the universality of the membrane traffic machinery across diverse eukaryotes. In the near future, our advanced imaging techniques, combined with biochemical, genetic, and pharmacological experiments, will further provide a wealth of new information to better understand the conserved molecular mechanisms underlying cargo traffic within the Golgi and between the Golgi and its neighboring organelles.

## Materials and methods
### Yeast strains and culture conditions
The yeast *S. cerevisiae* strains and plasmids used in this study are listed in *Supplementary files 2–4*. We used the yeast strain YPH499 (*Brachmann et al., 1998*) and the yeast GFP clone collection (parent strain, BY4741) (*Huh et al., 2003*). YPH499 *ADE2+* cells were generated by integration with pRS402 (*Brachmann et al., 1998*) digested by *Stu*I into the *ade2* site. For BFA experiments (*Figure 5*), we used the *erg6* mutant strain (3-E11) from the yeast deletion collection (parent strain, BY4741) (*Brachmann et al., 1998*), because wild-type strains of *S. cerevisiae* are insensitive to BFA and the *erg6* deletion renders them sensitivity to BFA (*Vogel et al., 1993*). The fluorescent proteins we used were GFP(S65T), EGFP, or yeast codon-optimized EGFP (yEGFP) for the green channel, mCherry or tagRFP for the red channel, and iRFP713 (*Filonov et al., 2011*) for the infrared channel. The insertion of the fluorescent protein gene into the yeast genome was performed by PCR-mediated gene replacement (*Janke et al., 2004*; *Longtine et al., 1998*) and verified by PCR and fluorescence microscopy. Plasmid-based constructs encoding fluorescent protein-tagged proteins of interest were expressed under the control of the *ADH1* promoter, the *TPI1* promoter, or their own promoter.

For microscopic observation, the yeast cells were grown in selective medium (0.67% yeast nitrogen base without amino acids and 2% glucose) with appropriate supplements. The cells were harvested at an early-to-mid logarithmic phase and then seeded onto glass coverslips coated with concanavalin A.

### SCLIM observation
For 3D and 4D imaging, the cells were observed by SCLIM developed by us (*Kurokawa and Nakano, 2020*; *Tojima et al., 2023*) at room temperature. The system consists of an inverted microscope (IX73; Evident/Olympus) equipped with solid-state lasers emitting at 473 nm (Blues, 50 mW; Cobolt), 561 nm (Jive, 50 mW; Cobolt), and 671 nm (CL671-100-S, 100 mW; CrystaLaser), a 100× objective (UPlanSApo, [NA 1.4] or UPlanXApo, [NA 1.45]; Evident/Olympus), a custom-built piezo actuator (Yokogawa Electric), a high-speed spinning-disk confocal scanner (CSU-10; Yokogawa Electric), a custom-built emission splitter unit, three image intensifiers (Hamamatsu Photonics) with custom-built cooling systems, and three EM-CCD cameras (ImagEM; Hamamatsu Photonics) for green, red, and infrared channels. For 3D (xyz) observations, 41 optical slices 0.2 µm apart (total z-range: 8 µm) were

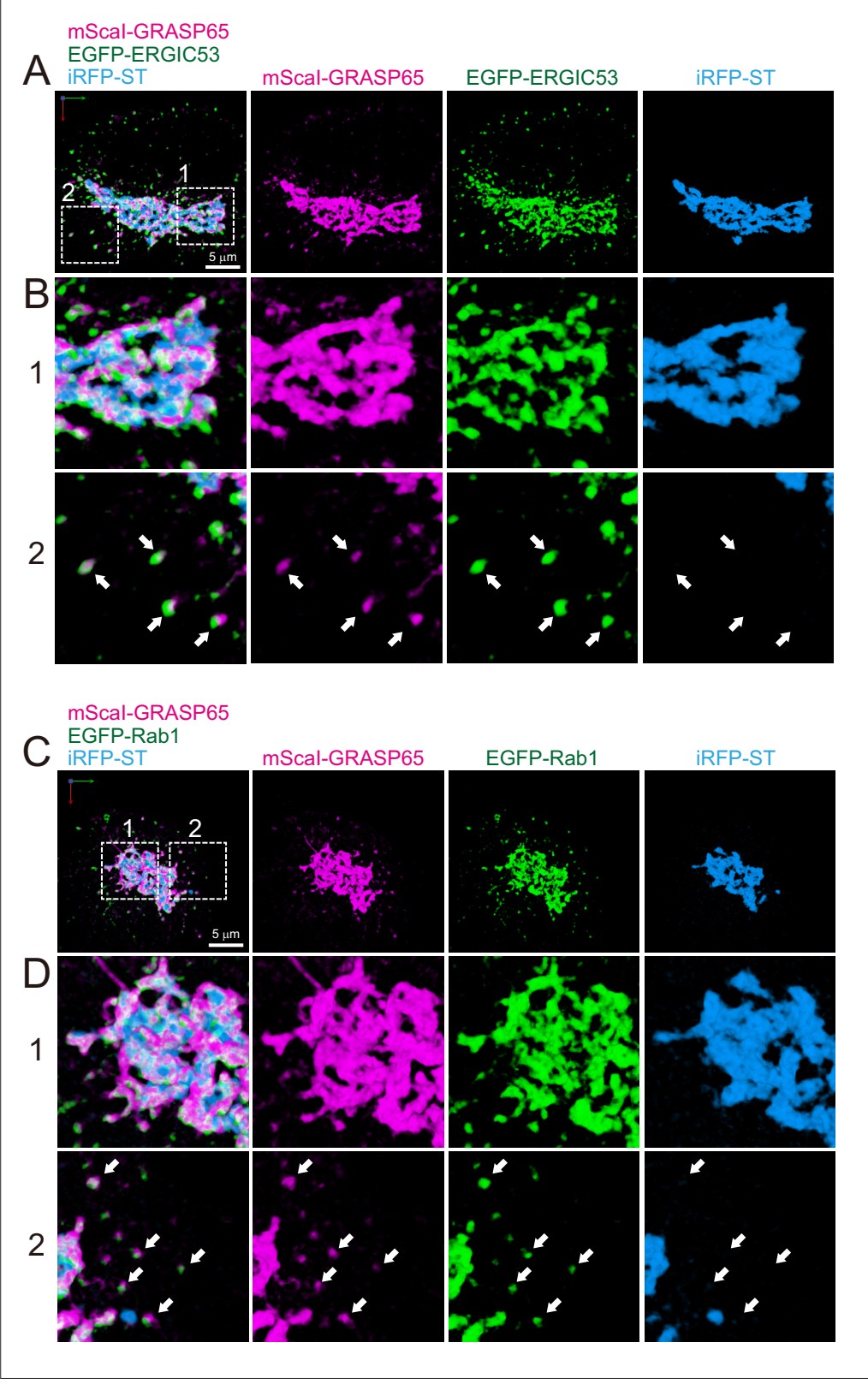

**Figure 11.** Distribution of endoplasmic reticulum-Golgi intermediate compartment (ERGIC) proteins (GRASP65, ERGIC53, and Rab1) in HeLa cells. (**A** and **B**) Triple-color three-dimensional (3D) super-resolution confocal live imaging microscopy (SCLIM) images of the Golgi ribbon area of a HeLa cell expressing mScarlet-I (mScaI)-GRASP65 (magenta), EGFP-ERGIC53 (green), and iRFP-ST (Golgi marker, blue). (**C** and **D**) Triple-color 3D SCLIM

*Figure 11 continued on next page*

*Figure 11 continued*

images of the Golgi ribbon area of a HeLa cell expressing mScaI-GRASP65 (magenta), EGFP-Rab1 (green), and iRFP-ST (blue). Scale bars: 5 µm. (**A** and **C**) Images of the whole Golgi ribbon. The left-most panels show merged images. The right panels show mScaI, EGFP, and iRFP fluorescence signals, respectively. (**B** and **D**) Zoom-up images in the white dashed squares (1 and 2) shown in (**A**) and (**C**). Arrows indicate that mScaI-GRASP65 and EGFP-ERGIC53/Rab1 double positive but iRFP-ST-negative puncta distributed away from the Golgi ribbon.

collected at 4–15 frames/s. Under most of experimental conditions of SCLIM employed in the present study, the spatial resolution in 2D is less than 180 nm and the temporal resolution is 1 volume per 5 s at most in 3D.

To observe cisternal maturation in 4D (xyz plus time) (*Figures 1–3, 6A–H, 7, 8A–H and K–O*), 21 optical slices 0.2 µm apart (total z-range: 4 µm) were acquired at 15 frames/s every 5 s. We selected cisternae of interest so that they could be tracked continuously and both red and green fluorescence peaks could be detected during the observation period. The mean peak-to-peak values with SD of all number of cisternae are shown in *Table 1*. To observe ERES-ERGIC interaction in 4D (*Figure 4D*), nine optical slices 0.25 µm apart (total z-range: 2 µm) were acquired at 20 frames/s every 1.1 s. Z-stack images were converted into 3D voxel data and subjected to deconvolution (fast or iterative resto-ration) using Volocity software (Perkin Elmer) with a theoretical point-spread function for spinning-disk confocal microscopy. The 3D images were visualized using the '3D opacity' mode of Volocity. The fluorescence intensities ($F$) for green, red, or far-red channels were averaged within the region of interest (ROI) that covers single cisterna. To normalize the fluorescence intensity, relative fluorescence over the peak fluorescence ($F/F_{peak}$) was calculated, where $F_{peak}$ was the maximum $F$ value during the observation period. For co-localization analysis, Pearson's correlation coefficient values were calcu-lated using Volocity. The ROI was set to cover an entire single cell and the signal threshold was set using the Costes' method (*Costes et al., 2004*).

## BFA experiments

BFA (Sigma, #B7651) was dissolved in DMSO to prepare a 50 mM stock solution. Aliquots of the stock solution were added to the culture media to a final concentration of 200 µM (0.4% DMSO) for the experiments. The same amount of DMSO alone was used as a vehicle control. The BFA/DMSO-treated cells were observed using an inverted epifluorescence microscope (IX83, Evident/Olympus) equipped with a 100× objective (UPlanXApo) and a sCMOS camera (ORCA-FusionBT, Hamamatsu Photonics) (*Figure 5A, B, and E–G*) or SCLIM (*Figure 5C and D*). Perfusion experiments (*Figure 5E–G*) were performed using a microfluidic perfusion chamber system (CellASIC ONIX2, Millipore) with a flow pressure of 2 psi at 20°C according to the manufacturer's protocol.

## Culture and transfection of HeLa cells

HeLa cells (RIKEN BioResource Research Center, #RCB0007) were cultured and transfected as described previously (*Tojima et al., 2023*). Cell line identity was verified by short tandem repeat profiling (Promega) and cells were negative for mycoplasma contamination (MycoStrip, Invivogen). Expression plasmids encoding EGFP-Sec16B, mCherry-KDEL, and EGFP-Rab1 were obtained from Addgene (#66607, #55041, and #49467 respectively). Plasmids encoding iRFP-ST, EGFP-ERGIC53, and mScarlet-I-GRASP65 under the control of CMV promoter were generated from mCherry-ST (Addgene, #55133), iRFP713 (Addgene, #31857), pMXs-IP spGFP-ERGIC53 (Addgene, #38270), EGFP-GRASP65 (Addgene, #137709), and mScarlet-I-Giantin (Addgene, #85050) using In-Fusion Cloning kit (Takara).

## Acknowledgements

We are grateful to Yoko Ito, Tomohiro Uemura, and Yutaro Shimizu, and all the members of the Live Cell Super-Resolution Imaging Research Team at RIKEN Center for Advanced Photonics for fruitful discussions on this manuscript. We also thank Miho Waga and Kalai Madhi Muniandy for making plasmid constructs and yeast strains, and Kumiko Ishii for microscopic observation of HeLa cells. This work was supported by Grants-in-Aid for Scientific Research from the Ministry of Education, Culture, Sports, Science, and Technology (MEXT) of Japan (grant numbers 19K06669, 19H04764, and

22K06213 to TT; 17H06420, 18H05275, and 23H00382 to AN) and the CREST program of the Japan Science and Technology Agency (JST) (grant number JPMJCR21E3 to TT).

## Additional information

### Funding

| Funder | Grant reference number | Author |
|---|---|---|
| Ministry of Education, Culture, Sports, Science and Technology | KAKENHI 19K06669 | Takuro Tojima |
| Ministry of Education, Culture, Sports, Science and Technology | KAKENHI 19H04764 | Takuro Tojima |
| Ministry of Education, Culture, Sports, Science and Technology | KAKENHI 22K06213 | Takuro Tojima |
| Japan Science and Technology Agency | CREST JPMJCR21E3 | Takuro Tojima |
| Ministry of Education, Culture, Sports, Science and Technology | KAKENHI 17H06420 | Akihiko Nakano |
| Ministry of Education, Culture, Sports, Science and Technology | KAKENHI 18H05275 | Akihiko Nakano |
| Ministry of Education, Culture, Sports, Science and Technology | KAKENHI 23H00382 | Akihiko Nakano |

The funders had no role in study design, data collection and interpretation, or the decision to submit the work for publication.

### Author contributions

Takuro Tojima, Conceptualization, Resources, Data curation, Formal analysis, Funding acquisition, Investigation, Visualization, Methodology, Writing – original draft; Yasuyuki Suda, Natsuko Jin, Kazuo Kurokawa, Resources, Validation, Methodology; Akihiko Nakano, Conceptualization, Supervision, Funding acquisition, Validation, Writing - review and editing

### Author ORCIDs

Takuro Tojima (iD) https://orcid.org/0000-0002-9140-3205
Yasuyuki Suda (iD) http://orcid.org/0000-0001-8725-8001
Natsuko Jin (iD) http://orcid.org/0000-0002-0231-3057
Kazuo Kurokawa (iD) http://orcid.org/0000-0003-3549-4795
Akihiko Nakano (iD) http://orcid.org/0000-0003-3635-548X

### Decision letter and Author response

Decision letter https://doi.org/10.7554/eLife.92900.sa1
Author response https://doi.org/10.7554/eLife.92900.sa2

## Additional files

### Supplementary files

• Supplementary file 1. Alignment of amino acid sequences of yeast Emp46/47 and human ERGIC-53.

• Supplementary file 2. Yeast strains used in this study.

• Supplementary file 3. Yeast plasmids used in this study.

- Supplementary file 4. Yeast strains and plasmids used in each experiment.
- MDAR checklist

## Data availability

Source data contains the numerical data used to generate the figures and table.

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
