## [Editor Report]

This paper provides important insights into the spatiotemporal mapping of a variety of proteins in and around the Golgi apparatus in budding yeast, with compelling evidence from high-resolution microscopy techniques. This research significantly advances our understanding of intracellular processes and represents a valuable contribution to the field of membrane trafficking and cell biology.

---

## [Decision Letter]

[Editors' note: this paper was reviewed by Review Commons.]

---

## [Author Response]

First of all, we thank all the reviewers for their careful reading and constructive comments on our manuscript. As suggested by the reviewers, we have performed several additional experiments and revised the manuscript. The new data are presented in Figure 1D, 1I, 1N, 1R-V, 2D, 2H, 2H, 3D, 3H, 3L, 3P, 4D, 6D, 6H, 7D, 7H, 7L, 7P, 7T, 8D, 8H, 8N, Figure 1–supplement 1 and 2, Figure 2–supplement 1, Figure 3–supplement 1, Figure 5–supplement 1, Figure 6–supplement 1, Figure 7–supplement 1, Figure 8– supplement 1 and 2, and Supplementary file 1. Also, Figure 9 diagram has been updated based on the new data.

Our general statements and point-by-point responses to the reviewers’ comments are as follows.

General statements

1. We have tried to revise the manuscript to meet the consensus of the three reviewers as much as possible, but there is one point of disagreement among them: regarding the late Golgi mapping data in the second half of the paper, Reviewer #1 says it should be toned down, while Reviewers #2 and #3 say it should be emphasized. We understand both opinions, but we have revised our manuscript to follow the latter in the current version. However, we can reconsider to conform to the former’s view, depending on the editor’s opinion.

2. Reviewers #1 and #3 made important comments regarding the definition of the ERGIC. We feel that the three reviewers seem to have different ideas about the concept of ERGIC. The confusion in the definition of ERGIC may make it controversial whether the yeast pre-Golgi compartment we discovered in the present study should be called ERGIC, GECCO, or part of the Golgi.

The ERGIC was originally identified in mammalian cells as a mobile carrier responsible for long-distance cargo transport from the ER exit site (ERES) at the cell periphery to the Golgi ribbon assembled near the centrosome. However, the cis-most compartment of the mammalian Golgi contains a set of ERGIC marker proteins. In plants, the ERES and the Golgi stack are adjacent, so there is no need for such long-distance carriers as in mammals. However, we have previously shown that the cis-most cisterna of the plant Golgi corresponds to the mammalian ERGIC and is a functionally specialized compartment for cargo reception, which we named GECCO (Ito et al., 2018. J Cell Sci. 131: jcs203893. [PMID: 28839076]; Ito et al., 2012. Mol Biol Cell. 23: 3203-14. [PMID: 22740633]). The yeast pre-Golgi compartment found in the present study well corresponds to mammalian ERGIC and plant GECCO in terms of molecular components, functions, and sensitivity to BFA, so we would like to propose in this paper that they are homologous organelles. Since each Golgi cisterna in yeast moves around in the cytoplasm away from the ERES, we believe that the new yeast compartment is more in line with the definition of ERGIC, which is based on the mobility as an ER-Golgi carrier. Therefore, we use the term “yeast ERGIC” in this paper, but we are also comfortable with calling it “yeast GECCO.”

Point-by-point responses to the ReviewersReviewer #1Major comment #1-1: The concept of the ERGIC in mammalian cells was initially proposed based on the protein ERGIC-53 in the 1990s. However, recent nanoscopy imaging data from the Lippincott-Schwartz lab challenges the conventional view of ERGIC by revealing the "ERGIC" is a membrane domain of the ERES (Wegel et al., Cell, 2021; PMID: 33852913), suggesting it might not be appropriate to adopt this concept.The authors' observations could be interpreted differently. Since the ERGIC is not molecularly defined in their study, the authors cannot prove its existence in yeast unequivocally. Their data indicate the presence of Golgi cisternae, characterized by Grh1, that precede the earliest known cisternae. Although the authors refer to these Grh1positive cisternae as the "ERGIC", they are essentially "pre-early" cisternae that progress to become the early Golgi cisternae. Nevertheless, their findings could extend the budding yeast Golgi cisternal progression unit further upstream to include the ERES as the starting point for Golgi cisternal maturation. To further explore this, it would be interesting to investigate the kinetics of COPII subunits in cisternal progression along with Grh1 or Mnn9 and to plot COPII components in the Figure 9 map.

As mentioned in General statement 2, we think this is a matter of definition. Since the new yeast pre-Golgi compartment found in the present study corresponds well to mammalian ERGIC and plant GECCO in terms of molecular components (e.g., ERGIC53/Emp46 and Rab1/Ypt1), functions (cargo reception from the ER), and sensitivity to BFA, we would like to define it as “yeast ERGIC” in this paper. At present, mammalian GRASP65 (yeast Grh1) is often considered as a marker of cis Golgi, but we found in the present study that Grh1 appeared prior to the ERGIC markers (Emp46 and Ypt1) in yeast. Therefore, we think that Grh1 should be defined as another ERGIC marker rather than a cis-Golgi marker, at least in yeast. We have explained this point in the revised text.

As suggested by the reviewer, we have performed dual-color 4D-SCLIM observations of Sec13 (an ERES marker) vs Grh1 (an ERGIC/GECCO marker) and found that some Grh1-positive puncta (ERGIC/GECCO) exhibited approach and contact (hug-and-kiss) behavior toward the Sec13-positive puncta (ERES). The new data are presented in Figure 4D, and explained in Abstract and text in the revised version. Because the hug-and-kiss behaviors do not fit well into the maturation time course map, we did not place Sec13 in Figure 9.

Major comment #1-2: The second half of the manuscript appears to deviate from the main focus on identifying the ERGIC. This section primarily presents the Golgi localization of four Golgi proteins (Ypt1, Gea1, Gea2, and Alp6) deduced from kinetics. However, it lacks functional studies to substantiate the authors' claims on their cellular functions. As a result, this part of the study remains purely speculative and might not support the authors' claims. Given that Figure 9 provides a highly informative summary of all kinetics and localization data, I recommend the authors keep but significantly abridge this section.

As mentioned in General statement 1, Reviewers #2 and #3 found the second half of the manuscript to be very important, and they feel that more emphasis should be placed on it. We understand both opinions, but we have revised the manuscript to follow the opinion from Reviewers #2 and #3 in the title, Abstract, and Introduction. However, we can reconsider to follow Reviewer #1’s view, depending on the editor’s opinion.

Major comment #1-3: The analysis of only one fluorescent particle or Golgi cisternal punctate structure is insufficient for a Golgi marker, considering the substantial variation of Golgi cisternae. To improve statistical robustness, the authors should select multiple fluorescent particles from multiple cells, displaying plots with averaged intensities, error bars, and sample sizes (n).

In the original version, statistical information (n and mean peak-to-peak time with SD) was provided only in supplementary material. In the revised version, we have moved this to the main table (Table 1). Also, we have reworked these data as graphs (Figure 1–supplement 1F, Figure 2–supplement 1D, Figure 3– supplement 1E, Figure 6–supplement 1C, Figure 7–supplement 1F, and Figure 8– supplement 1D). Furthermore, we have shown the averaged time course changes in the normalized fluorescence intensities of the green and red channels during maturation (mean ± SEM) (Figure 1D, 1I, 1N, 1R, 1V, 2D, 2H, 2L, 3D, 3H, 3L, 3P, 6D, 6H, 7D, 7H, 7L, 7P, 7T, 8D, 8H, and 8N), and those of representative 6 individual data (Figure 1– supplement 1A-E, Figure 2–supplement 1A-C, Figure 3–supplement 1A-D, Figure 6–supplement 1A,B, Figure 7–supplement 1A-E, and Figure 8–supplement 1A-C).

Major comment #1-4: In Fig. 5A, the BFA-induced lumps positive for Grh1, Rer1, and Sed5 may potentially represent the ERES, as observed in mammalian cells (Ward et al., JCB, 2002; PMID: 11706049). To verify this, the authors should co-label these lumps with COPII subunits.

As suggested, we have performed an additional experiment examining the effect of BFA on ERES distribution by SCLIM. Upon treatment with BFA, the number of ERES (Sec13positive puncta) did not change, while Rer1-positive compartments formed small numbers of larger structures (GECCO). This indicates that BFA had no effect on the ERES distribution. The new data are presented in Figure 5–supplement 1 and text in the revised version.

Major comment #1-5: The authors previously reported the "hug-and-kiss" model for cargo transport from the ERES to the early Golgi cisternae. As the current study is highly relevant to the "hugand-kiss" model, it is disappointing that the authors did not provide further data and comment on it. The "hug-and-kiss" and "ERGIC" transport modes are two distinct ways for secretory cargo transport from the ERES to the early Golgi cisternae. The authors should verify the "hug-and-kiss" transport and report the relative frequency of the two transport modes.

As mentioned above (Response to Major comment #1-1), we have performed dual-color 4D-SCLIM observations of Sec13 vs Grh1, and found that some Grh1-positive puncta (ERGIC/GECCO) exhibited approach and contact (hug-and-kiss) behavior toward the Sec13-positive puncta (ERES). On the other hand, we hardly detected ERGIC formation from the ERES, suggesting that cargo transport by hug-and-kiss action is dominant. The new data are presented in Figure 4D, Abstract, and text in the revised version.

Major comment #1-6: The current version has minimal background knowledge of ERGIC in mammalian and yeast cells. Therefore, the authors should provide a comprehensive introduction to ERGIC.

We have added some background knowledge of mammalian ERGIC and plant GECCO in the revised text.

Reviewer #2Major comment #2-1: There is an enormous amount of patient systematic analysis packed in this paper, following a punctate organelle as it emerges from the dark, evolves over time (following a combination of 2 markers) with fluorescence peaks at specific time points, after which the signal disappears again. I am certain other cell biologists will be impressed, as I was, viewing the individual images and graphs presented, culminating ultimately in figure 9 that could go straight into a textbook to form a starting point for anybody who wishes to study a particular protein of interest and chose the most appropriate markers to compare it with.The authors propose that the ERGIC/GECCO/Golgi-remnants compartment is an evolutionary conserved structure even though it has a different subcellular distribution/morphology in different classes of eukaryotes. The data presented here and in earlier work seem to support this notion. In particular, the authors demonstrate that the yeast GRASP 65 homologue Grh1 is the earliest to appear closely followed by Ypt1 and Emp46. The fact that RER1 and ERD2 come slightly later is in line with a proposed gate-keeper function, because if they were instead to recycle continuously they should appear first in line. I agree with the authors that the simple model of ER-derived COPII vesicles fusing with each other and thus creating an ERGIC/GECCO de novo is probably too simplistic. The idea of a more permanent structure, pulsating between cargo-loading and cargo-releasing events, possibly associated with creating zones/subdomains within a single cisterna seems very attractive given the data shown here.This work is descriptive, but it is of very high importance to anybody engaged with experimental approaches to study protein sorting from the Golgi-apparatus back to the ER, or on to the plasma membrane or the lytic compartments. The 5 functional stages proposed for Golgi-maturation is an attractive starting point for future research, and I very much like the notion that ERGIC and cis-Golgi cisternae may start as zones/subdomains within a single cisternae, possibly formed via phase separations involving both proteinprotein and protein-lipid interactions.

We appreciate very warm and favorable comments from this reviewer. We are very much encouraged.

Minor comment #2-1: The title strongly focusses on the ERGIC and therefore the earliest sorting steps in the ER-Golgi system, but this manuscripts offers so much more. I was fascinated to learn that Ypt1 appears twice during cisternal maturation in yeast. This may be a yeast-specific phenomenon but it is very interesting. The same can be said about the proposal that Gea1 and Gea2 have different roles in the Golgi and act in different cisternae, and the localisation of AP-3 at the trans-Golgi rather than the TGN. The functional distinction between trans-Golgi and TGN and the differences in their origin are important points and it will be a shame if readers don't realise that this manuscript offers further insight into later steps in Golgi-mediated transport. There may be a case to add something to the abstract and/or modify the title accordingly, but then I also feel that long titles are not ideal and the ERGIC/GECCO portion is the more important take-home message. This is a case for the editorial team and the authors to make the most of the findings.

As we mentioned in General statement 1, we have revised our manuscript to follow the opinion of Reviewers #2 and #3. As suggested, we have changed the title to emphasize our findings on the late Golgi: "Spatiotemporal dissection of the Golgi apparatus and the ER-Golgi intermediate compartment in budding yeast". We have also changed the condensed title to “Mapping of ERGIC-Golgi-TGN maturation in yeast”. Abstract and Introduction have been revised accordingly.

Minor comment #2-2: Given the importance of ERD2 in sorting soluble proteins to be returned back to the ER, the authors may consider using the biological active XFP-TM-ERD2 fusion instead of ERD2-GFP, but this may be kept for future work. In plants, ERD2-GFP is mainly at the Golgi when overexpressed, its erroneous leakage to the ER is only observed at low expression or when K/HDEL proteins are co-expressed. The XFP-TM-ERD2 construct may be better confined to the ERGIC-GECCO and may have a different temporal pattern.

Thank you very much for suggesting an important experiment to confirm the Erd2 dynamics. As the reviewer pointed out, the C-terminal tagging of plant ERD2 with XFP causes loss of function and mislocalization in plants (Alvim et al., 2023. Nat Commun. 14:1612. [PMID: 36959220]; Silva-Alvim et al., 2018. Plant Cell. 30:2174-2196. [PMID: 30072420]). However, we use a yeast strain in which GFP is integrated into the genome at the C-terminus of the Erd2 gene (BY4741 ERD2-GFP::HIS3MX6; Huh et al., 2003. Nature. 425:686-69. [PMID: 145620951]). We confirmed that this strain is viable and grows normally (Author response image 1) and has been used successfully by another group previously (e.g., Geva et al., 2017.Traffic. 18:672-682. [PMID: 28727280]). Considering that Erd2 is an essential gene, these results strongly suggest that, at least in yeast, the Erd2 function is maintained even when it is GFP-tagged at the C-terminus. We agree that the use of XFP-TMD-ERD2 would be an interesting experiment. However, since the TMD portion of this construct is derived from ERP1, an ERD2-related protein that is found only in plants (but not in yeast), we need to start with basic assays to verify that this construct functions as endogenous Erd2 in yeast cells, which would be time consuming. In addition, this experiment seems to be a bit too specific from the viewpoint of the main theme of this paper, which is to map a large number of ERGIC/Golgi/TGN proteins. For these reasons, we would like to skip this experiment in the present work.

**Author response image 1. sa2fig1:** C-terminal GFP tagging of Erd2 in genome had no effect on yeast growth. Growth assay of serially diluted yeast cells. Cells were incubated overnight at 30°C on a YPD plate.

Minor comment #2-3: The authors may consider citing Stornaiuolo et al., Mol Biol Cell 2003 Mar;14(3):889-902 who compared the trafficking of KDEL and KKXX pathways and concluded that KDEL proteins are retrieved prior to KKXX proteins….as this fits nicely into the current findings showing that ERD2 and RER1 appear sooner than COPI markers.

Thanks for sharing information about this important paper with us. We have cited it in the revised text.

Reviewer #3Major comment #3-1: It is not clear to me how the presented data shows the existence of an ERGIC in the yeast *S. cerevisiae*. I understand, and appreciate from this text too, that a clear definition of ERGIC, even in a mammalian system, is unclear. For this reason, I would first suggest that the authors provide a clear definition of what ERGIC means to them. Next, the experiments herein presented are all based on a very careful, thorough and nicely organized spatio-temporal mapping of a large number of early secretory pathway proteins, including the ERGIC53 yeast "counterpart" Emp46 (it would help to add, even as a supplementary figure, an alignment/sequence comparison between the human ERGIC53 and the S. cerevisiae emp46). However, the data presented here does not clearly indicate to me that there is a bona fide ERGIC in yeast. Couldn't it just be that what the authors call ERGIC is a cis-Golgi cisterna? I understand that the BFA experiments show a different behavior for some proteins, which fits with what the authors previously names GECCO in plants, so why not calling this GECCO? Again, it will be important to provide definitions of these compartments for the audience.

We totally agree that the definition of ERGIC is elusive and many researchers have different ideas about the concept of ERGIC, even in the mammalian system. However, we think the yeast pre-Golgi compartment found in the present study corresponds well to mammalian ERGIC and plant GECCO in terms of molecular components (e.g., ERGIC53/Emp46 and Rab1/Ypt1), functions (e.g., cargo receiver and carrier), and sensitivity to BFA. Since each Golgi cisterna in yeast moves around in the cytoplasm away from the ERES, we feel that the new yeast compartment is more in line with the original concept of mammalian ERGIC, which is based on the mobility as an ER-Golgi carrier. Thus, we define it as “yeast ERGIC” in this paper. We have added some explanation on this point in the revised version.

We have shown the alignment data of mammalian ERGIC-53 vs yeast Emp46/47 in the revised version (Supplementary file 1) .

Major comment #3-2: Next, my main concern here is that this is all based on SCLIM, which is a very nice technique, but the resolution is limited in both space and time (by the way, it would be nice to explicitly measure of quantify the spatial resolution in x-y-z). Hence, it is not possible to discern whether an "independent" ERGIC is formed as compared to cis-Golgi cisterna. Electron microscopy (possibly CLEM) could help somehow resolve that and massively increase the strength of the claims, but I do understand this might be difficult for this group and very time consuming, so it might be important to clearly state the limitation of the herein presented data. A possible alternative to test if protein that are seen segregated are within the same membrane (as claimed here) would be to do trapping experiments where a reagent induces dimerization between the two proteins (when tagged with specific tags, such as FKBP/FRB).

As the reviewer mentioned, CLEM analysis is not easy for us to perform in a timely fashion. We have previously performed 3D CLEM analysis through a fruitful collaboration with EM experts and shown that Golgi and TGN markers are segregated within a continuous membrane compartment (Kurokawa et al., 2019. J Cell Biol. 218:1602-1618. [PMID: 30858192]). We have added this information in the revised version. Regarding the spatial resolution of SCLIM, we have added some information.

To try to address the problem the reviewer suggested, we performed the FKBP/FRB-induced trapping experiments (Author response image 2). We made rapamycin-resistant yeast strain *TOR1-1* and *fpr1∆* in which FRB-GFP is fused to Grh1 (an ERGIC protein), FKBP is fused to Pma1 (a plasma membrane protein), and red fluorescent protein (mCherry or mRFP) is fused to Sed5 (another ERGIC protein), Mnn9 (a cis Golgi protein), Sec21 (a medial Golgi protein), Sec7 (a TGN protein), or Sec13 (an ERES protein), to test whether or not they are located within the same membrane compartment and move together. As a control experiment, we tested Grh1-mCherry. Upon treatment with rapamycin, the Grh1-FRB-GFP-containing compartments were successfully trapped at the plasma membrane where Pma1 is located. However, in this situation, none of these target proteins examined, including control (Grh1-mCherry), were targeted to the plasma membrane together with Grh1-FRB-GFP. These results suggest that, at least in our experimental system, when Grh1-FRB-GFP is captured at the plasma membrane by rapamycin, the membrane compartment (cisterna) on which Grh1 rode does not migrate together with it. In addition, the time scale of Golgi cisternal maturation in yeast is only a few minutes, whereas the assembly of Grh1-FRB-GFP at the plasma membrane after rapamycin treatment takes about 30 minutes. Thus, this type of experiment appears to be inappropriate in terms of time-scales. Therefore, we did not include these results in the revised text.

**Author response image 2. sa2fig2:** FKBP12/FRB-induced trapping experiments. (A) The strategy of the experiments. We made rapamycin-resistant yeast strains in which FRB-GFP is fused to Grh1 (an ERGIC protein), FKBP12 is fused to Pma1 (a plasma membrane protein), and red fluorescent protein (mCherry or mRFP) is fused to a target protein. If the Grh1-FRB-GFP is recruited to the plasma membrane (where Pma1-FKBP12 is localized) together with the target protein after treatment with rapamycin, this suggests that they were in the same membrane compartment. (B) Control experiment. Upon treatment with rapamycin, Grh1-FRB-GFP was recruited to the plasma membrane, but Grh1-mCherry was not. (C-G) Sed5 (another ERGIC protein), Mnn9 (a cis-Golgi protein), Sec21 (a medial Golgi protein), Sec7 (a TGN protein), and Sec13 (an ERES protein) were also not recruited to the plasma membrane. These results suggest that, at least in this experimental system, when Grh1-FRB-GFP is captured at the plasma membrane by rapamycin treatment, the membrane compartment on which Grh1 rode does not migrate together with it.

Major comment #3-3: I could not find any details (maybe I have missed them) about how many times experiments were replicated and the statistical significance of the findings herein reported. In most figures, examples of microscopy images/videos are shown, and selected lines profiles are presented. However, it is not clear how robust these experiments are. Some ideas:(2.1.) The major source of quantification is the peak-to-peak time distance between two proteins. In Table S1 some stdev is presented, but not clear how it is find it is the sted of all n number of puncta? or of the mean duration per cell? or of the mean duration per experiment? I would suggest that the authors provide the results shown in Table S1 plotted as a histogram or superplot see e.g. https://rupress.org/jcb/article/219/6/e202001064/151717/SuperPlots-Communicatingreproducibility- and clearly explain how statistics is performed.

As suggested, we have reworked the peak-to-peak time data as graphs (Figure 1– supplement 1F, Figure 2–supplement 1D, Figure 3–supplement 1E, Figure 6– supplement 1C, Figure 7–supplement 1F, and Figure 8–supplement 1D). In addition, we have presented the averaged time course changes in normalized fluorescence intensities of green and red channels (mean ± SEM) (Figure 1D, 1I, 1N, 1R, 1V, 2D, 2H, 2L, 3D, 3H, 3L, 3P, 6D, 6H, 7D, 7H, 7L, 7P, 7T, 8D, 8H, and 8N), and those of representative 6 individual data (Figure 1–supplement 1A-E, Figure 2–supplement 1A-C, Figure 3–supplement 1A-D, Figure 6–supplement 1A, B, Figure 7– supplement 1A-E, and Figure 8–supplement 1A-C). We have also added explanations about statistics performed in the revised text and figure legends. SD values in Table 1 (Table S1 in the original version) is those of all number of puncta.

Major comment #3-4: (2.2) Also, the time-lapse movies are acquired with a 5s gap between time points. How is this included in the incertainty of the peak-to-peak duration in Table S1?

Individual peak-to-peak time data are discrete, every 5 s. We used the data as they are, including the uncertainty. We have presented all the individual data as scatter plots with mean ± SD (Figure 1–supplement 1F, Figure 2–supplement 1D, Figure 3– supplement 1E, Figure 6–supplement 1C, Figure 7–supplement 1F, and Figure 8– supplement 1D).

Major comment #3-5: (2.3) In pg. 7 the authors write "Although experimental variation was high, the two zones appeared to be spatially segregated". Can the authors provide quantitative and statistical support of this claim?

To show the experimental variation, we have presented many line-profile data in the revised version (Figure 1–supplement 2 and Figure 8–supplement 2). In Figure 1– supplement 2, many cisternae showed segregation patterns of Emp46-positive zone and Mnn9-positive zone, while some others showed overlapping distribution patterns. In Figure 8–supplement 2, all the cisternae examined showed clear segregation patterns of Apl6-positive zone and Sys1-positive zone. We have also changed the descriptions in the revised text.

Major comment #3-6: (Figure 9: the arrows should go from protein2.4) It is not clear to me how the puncta for analysis are selected. For example, in Figure 1C, the punctum shown already shows some initial co-localization (it could be e.g. that a peak value was prior or after the duration of the time lapse movie, thereby biassing the computation of the peak-to-peak duration). So, if one would consider those spots e.g., positive for Emp46 that do not contain Mnn9 signal, how often do you see conversion (that is, appearance of Mnn9 signal)? Along the same lines, in pg. 8 the authors write "… signal appeared first and then mnn9-mCherry came up". Details on how this quantification is done and statistical analysis would be needed, to my opinion, to support the claim.

To examine the cisternal maturation, we selected cisternae of interest so that they could be tracked continuously and had both red and green fluorescence peaks could be detected during the observation period. It is not easy to quantify exactly how often the Emp46-positive puncta converted to Mnn9-positive ones, because ERGIC/Golgi cisternae in yeast are moving around randomly at high speed, often making it impossible to follow individual cisternae for a long period of time. In addition, two cisternae sometimes came so close that they appeared to overlap even at the resolution of SCLIM. Furthermore, to reduce photobleaching and time-lapse intervals, we acquired 21 XYoptical slices at 0.2 μm apart (total Z-range: 4 m) for a 3D image, which does not cover the entire thickness of yeast cells, resulting in some cisternae to move outside the field of view. Such cisternae were excluded from the analysis, and only a few cisternae were usable to track the entire maturation process for analysis. We have added some explanation on this point in the revised text.

Minor comment #3-2: Pg. 8: have the authors tested Rer1 vs Emp46?

We have performed the suggested experiment in the revised version. Since we previously found that mCherry-tagged Rer1 mislocalizes to the vacuole, we examined the dynamics of iRFP-tagged Rer1 together with GFP-tagged Emp46 for dual-color 4DSCLIM analysis. The new results have been presented in Figure 1S-V, Figure 1– supplement 1E, F and the revised text. We also included the peak-to-peak time data in Figure 9 and Table 1.

Minor comment #3-3: Pg. 8: I was of the impression that GRASP65 (GORASP1) is considered to be a cisGolgi protein (see e.g., Tie et al. eLife 2018). Then, what the authors call "ERGIC" couldn't it simply be a cis-Golgi cisterna?

As we mentioned in “General statement 2”, this would be a matter of definition. We agree that mammalian GRASP65 (yeast Grh1) is currently considered by many researchers as a marker of the cis-Golgi, but we found in the present study that Grh1 appears prior to the ERGIC markers Emp46 (mammalian ERGIC-53) and Ypt1 (mammalian Rab1) in yeast. Thus, we would like to propose in this paper that the cis-most cisterna of the mammalian Golgi that harbors GRASP65 corresponds to ERGIC/GECCO. We have added our view on the definition of ERGIC/GECCO across species in the revised version.

Minor comment #3-5: The first part of the manuscript (up to mid page 13) is clearly focused on defining ERGIC in yeast, then the paper appears as a set of experiments aimed at adding more components in their spatio-temporal mapping. This is ok, but is should be clearly motivated and explained in the Title, abstract and intro.

We have changed the title and modified Abstract and Introduction to emphasize the second part of the manuscript.

Minor comment #3-6: The visualization of colocalization according to the opacity (as said in the methods) is somehow confusing to me. Are the 3D images projections or 3D renderings (no axes are seen)? In e.g. Figure 6G or 8L, regions where green and magenta (or green and red) are colocalized do not appear white (or yellow), which visually suggests to the inattentive reader that there is no colocalization, when there is.

“3D opacity” is a display mode of a commercial software “Volocity”. We think this mode is the best to express 3D information and is good because merge areas appear yellow or white to some extent. In the revised version, for clarity, we have added several line scan data (Figure 1–supplement 2 and Figure 8–supplement 2). We have also added axis labels in the revised version (Figure 1E, 1J, and 8O).